# Power analysis of knockoff filters for correlated designs

**Jingbo Liu**
Institute for Data, Systems, and Society
Massachusetts Institute of Technology
Cambridge, MA 02139
jingbo@mit.edu

**Philippe Rigollet**
Department of Mathematics
Massachusetts Institute of Technology
Cambridge, MA 02139
rigollet@math.mit.edu

## Abstract

The knockoff filter introduced by Barber and Candès 2016 is an elegant framework for controlling the false discovery rate in variable selection. While empirical results indicate that this methodology is not too conservative, there is no conclusive theoretical result on its power. When the predictors are i.i.d. Gaussian, it is known that as the signal to noise ratio tend to infinity, the knockoff filter is consistent in the sense that one can make FDR go to 0 and power go to 1 simultaneously. In this work we study the case where the predictors have a general covariance matrix $\Sigma$. We introduce a simple functional called *effective signal deficiency (ESD)* of the covariance matrix of the predictors that predicts consistency of various variable selection methods. In particular, ESD reveals that the structure of the precision matrix plays a central role in consistency and therefore, so does the conditional independence structure of the predictors. To leverage this connection, we introduce *Conditional Independence knockoff*, a simple procedure that is able to compete with the more sophisticated knockoff filters and that is defined when the predictors obey a Gaussian tree graphical models (or when the graph is sufficiently sparse). Our theoretical results are supported by numerical evidence on synthetic data.

## 1   Introduction

Variable selection is a cornerstone of modern high-dimensional statistics and, more generally, of data-driven scientific discovery. Examples include selecting a few genes correlated to the incidence of a certain disease, or discovering a number of demographic attributes correlated to crime rates.

A fruitful theoretical framework to study this question is the linear regression model in which we observe $n$ independent copies of the pair $(X, Y) \in \mathbb{R}^p \times \mathbb{R}$ such that

$$Y = X^\top \theta + \xi \,,$$

where $\theta \in \mathbb{R}^p$ is an unknown vector of coefficients, and $\xi \sim \mathcal{N}(0, n\sigma^2)$ is a noise random variable. Throughout this work we assume that $X \sim \mathcal{N}(\mathbf{0}, \Sigma)$ for some known covariance matrix $\Sigma$. Note that for notational simplicity our linear regression model is multiplied by $\sqrt{n}$ compared to standard scaling in high-dimensional linear regression [BRT09]. Clearly, this scaling, also employed in [JM14] has no effect on our results. In this work, we consider asymptotics where $n/p \to \delta$ is fixed.

In this model, a variable selection procedure is a sequence of test statistics $\psi_1, \ldots, \psi_p \in \{0, 1\}$ for each of the hypothesis testing problem

$$H_0^{(j)} : \theta_j = 0, \qquad \text{vs.} \qquad H_1^{(j)} : \theta_j \neq 0, j = 1 \ldots, p \qquad (1)$$

When $p$ is large, a simultaneous control of all the type I errors leads to overly conservative procedures that impedes statistical significant variables, and ultimately, scientific discovery. The False Discovery

Rate (FDR) is a less conservative alternative to global type I error. The FDR of a procedure $(\psi_1, \ldots, \psi_p)$ is the expected proportion of erroneoulsy rejected tests. Formally

$$\text{FDR} := \mathbb{E}\Big[\frac{\#\{j \,:\, \psi_j = 1, \theta_j = 0\}}{\#\{j \,:\, \psi_j = 1\} \vee 1}\Big]$$

Since its introduction more than two decades ago, various procedures have been developed to provably control this quantity under various assumptions. Central among these is the Benjamini-Hochberg procedure which is guaranteed to lead to a desired FDR control under the assumption that the design matrix $\mathbf{X} = (X_1, \ldots, X_n)^\top \in \mathbb{R}^{n \times p}$ formed by the concatenation of the $n$ column vectors $X_1, \ldots, X_n$ is deterministic and orthogonal [BH95, STS04].

In the presence of correlation between the variables, that is when the design matrix fails to be orthogonal, the problem becomes much more difficult. Indeed, if the variables $X_j$ and $X_k$ are highly correlated, any standard procedure will tend to output a similar coefficient for both, or in the case of Lasso for example, simply chose one of the two variables rather than both.

Recently, the knockoff filter of Barber and Candès [BC15, CFJL18] has emerged as a competitive alternative to the Benjamini-Hochberg procedure for FDR control in the presence of correlated variables, and has demonstrated great empirical success [KS19, SKB$^+$]. The terminology "knockoffs" refers to a vector $\tilde{X} \in \mathbb{R}^p$ that is easy to mistake for the original vector $X$ but is crucially independent of $Y$ given $X$. Formally, $\tilde{X}$ is a *knockoff* of $X$ if (i) $\tilde{X}$ is independent of $Y$ given $X$ and (ii) for any $S \subset \{1, \ldots, p\}$, it holds

$$(X, \tilde{X})_{\mathsf{swap}(S)} \overset{d}{=} (X, \tilde{X}) \tag{2}$$

where $\overset{d}{=}$ denotes equality in distribution and $(X, \tilde{X})_{\mathsf{swap}(S)}$ is the vector $Z \in \mathbb{R}^{2p}$ with $j$th coordinate given by

$$Z_j = \begin{cases} X_j & \text{if } j \in (\{1, \ldots, p\} \setminus S) \cup (S + \{p\}) \\ \tilde{X}_j & \text{if } j \in S \cup (\{p+1, \ldots, 2p\} \setminus (S + \{p\})) \end{cases}$$

In words, for any vector $\mathbb{R}^{2p}$, the operator $(\cdot)_{\mathsf{swap}(S)}$ swaps each coordinate in $j \in S$ with the coordinate $j + p$ and leaves the other coordinates unchanged. We call a *knockoff mechanism* any probability family of probability distributions $(P_x, x \in \mathbb{R}^p)$ over $\mathbb{R}^p$ such that $\tilde{X} \sim P_X$ is a knockoff of $X$. Since the knockoff is constructed independently of $Y$, it serves as a benchmark to evaluate how much of the coefficient of a certain variable is due to its correlation with $Y$ and how much of it is due to its correlation with the other variables.

With this idea in mind, the knockoff filter is then constructed from the following four steps:

1. **Generate knockoffs.** For $i = 1, \ldots, n$, given $X_i \in \mathbb{R}^p$, generate knockoff $\tilde{X}_i \sim P_{X_i}$ and form the $n \times 2p$ design matrix $[\mathbf{X}, \tilde{\mathbf{X}}]$ where $\tilde{\mathbf{X}} = (\tilde{X}_1, \ldots, \tilde{X}_n)^\top \in \mathbb{R}^{n \times p}$ is obtained by concatenating the knockoff vectors.

2. **Collect scores for each variable.** Define the $2p$ dimensional vector [1] $\hat{\underline{\theta}}$ as the Lasso estimator

$$\hat{\underline{\theta}} = \underset{\underline{\theta} \in \mathbb{R}^{2p}}{\text{argmin}} \, \frac{1}{2n} \|\mathbf{Y} - [\mathbf{X}, \tilde{\mathbf{X}}]\underline{\theta}\|_2^2 + \lambda\|\underline{\theta}\|_1 \,, \tag{3}$$

   where $\mathbf{Y} = (Y_1, \ldots, Y_n)^\top$ is the response vector and, collect the differences of absolute coefficients between variables and knockoffs into a set $\mathcal{D} = \{|\Delta_j|, j = 1, \ldots, p\} \setminus \{0\}$ where $\Delta_j$'s are any constructed statistics satisfying certain symmetry conditions [BC15]. A frequent choice is

$$\Delta_j := |\hat{\underline{\theta}}_j| - |\hat{\underline{\theta}}_{j+p}|, j = 1, \ldots, p.$$

   In this work we replace $\hat{\underline{\theta}}$ by the debiased version $\hat{\theta}^u$ (see (7) ahead) in the above definition.

3. **Threshold.** Given a desired FDR bound $q \in (0, 1)$, define the threshold

$$T := \min\Big\{t \in \mathcal{D} \colon \frac{\#\{j \colon \Delta_j \leq -t\}}{\#\{j \colon \Delta_j \geq t\} \vee 1} \leq q\Big\}.$$

4. **Test.** For all $j = 1, \ldots, p$, answer the hypothesis testing problem (1) with test

$$\Psi_j = \mathbb{1}\{\Delta_j \geq T\}.$$

This procedure is guaranteed to satisfy FDR $\leq q$ [BC15, Theorem 1] no matter the choice of knockoffs. Clearly, $\tilde{X} = X$ is a valid choice for knockoffs but it will inevitably lead to no discoveries. The ability of a variable selection procedure $(\psi_1, \ldots, \psi_p)$ to discover true positive is captured its *power* (or true positive proportion) defined as

$$\text{PWR} = \mathbb{E}\left[\frac{\#\{j \,:\, \psi_j = 1, \theta_j \neq 0\}}{\#\{j \,:\, \theta_j \neq 0\}}\right]$$

Intuitively, to maximize power, knockoffs should be as uncorrelated with $X$ as possible while satisfying the exchangeability property (2). Following this principle, various knockoff mechanisms have been proposed in different settings, which typically involves solving an optimization to minimize a heuristic notion of correlation [BC15, CFJL18, RSC18]. Because of this optimization problem, knockoff mechanisms with analytical expressions are rare, with the exception of the equi-knockoff [BC15] and metropolized knockoff sampling [BCJW19]). Partly due to this, the theoretical analysis of the power of the knockoff filter has been very limited, even in the Gaussian setting. In the special case where $X \sim \mathcal{N}(0, D)$ for some *diagonal* matrix, i.e. when the variables are independent, one can simply take $\tilde{X} \sim \mathcal{N}(0, D)$ independent of $X$. In this case, the power of the knockoff filter tends to 1 as the signal-to-noise ratio tends to infinity [WBC17].

When predictors are correlated, [FDLL19] proved a lower bound on the power, where the limiting power as $n \to \infty$ is bounded below in terms of the number $p$ of predictors and extremal eigenvalues of the covariance matrix of the true and knockoff variables. While this lower bound provides a sufficient condition for situations when the power tends to 1, it is loose in certain scenarios. For example, if all predictors are independent except that two of them are almost surely equal, then the minimum eigenvalue of the covariance matrix is zero and yet, experimental results indicate that the FDR and the power of the knockoff filter are almost unchanged.

**Our contribution.** In this paper, we revisit the statistical performance of the knockoff filter $X \sim \mathcal{N}(0, \boldsymbol{\Sigma})$ and characterize the situation the knockoff filter is *consistent*, that is when its FDR tends to 0 and its power tends to 1 simultaneously. More specifically, under suitable limit assumptions, we show that the knockoff filter is consistent if and only if the empirical distribution of the diagonal elements of the precision matrix of $\underline{\mathbf{P}} := \underline{\boldsymbol{\Sigma}}^{-1}$ converges to 0, where $\underline{\boldsymbol{\Sigma}}$ denotes the covariance matrix of $[X, \tilde{X}] \in \mathbb{R}^{2p}$ converges to a point mass at 0. In turn, we propose an explicit criterion, called *effective signal deficiency* defined formally in (8) to practically evaluate consistency or lack thereof. Here the term "signal" refers to the covariance structure $\boldsymbol{\Sigma}$ of $X$ and the effective signal deficiency essentially how much weak such a signal should be for a knockoff mechanism to be consistent.

A second contribution is to propose a new knockoffs mechanism, called *Conditionally Independent Knockoffs* (CIK), which possesses both simple analytic expressions and excellent experimental performance. CIK does not exist for all $\boldsymbol{\Sigma}$, but we show its existence for tree graphical models or other sufficiently sparse graphs. Note that in practice, the so-called model-X knockoff filter requires the knowledge of $\boldsymbol{\Sigma}$, an estimation of which is often prohibitive except when the graph has sparse or tree structures. CIK has simple explicit expressions of the effective signal deficiency for tree models, since the empirical distribution of the diagonals of $\boldsymbol{\Sigma}^{-1}$ is the same as that of $(\mathbf{P}_{jj}^2 \boldsymbol{\Sigma}_{jj})_{j=1}^p$. We remark that CIK is different than *metropolized knockoff sampling* studied in [BCJW19] (originally appeared in [CFJL18, Section 3.4.1]), even in the case of Gaussian Markov chains. The latter exists for generic distributions and is computationally efficient for Markov chains.

**Notation.** We write $[n] := \{1, \ldots, n\}$ and $\mathbf{1}$ to denote the all-ones vector. For any vector $\theta$, let $\|\theta\|_0$ and $\|\theta\|_1$ denote its $\ell_0$ and $\ell_1$ norms. Given a vector $\mathbf{x}$, we denote by $\text{diag}(\mathbf{x})$ the diagonal matrix whose diagonal elements are given by the entries of $\mathbf{x}$ and for a matrix $\mathbf{M}$, we denote by $\text{diag}(\mathbf{M})$ the vector whose entries are given by the diagonal entries of $\mathbf{M}$. For a standard Gaussian random variable $\xi \sim \mathcal{N}(0, 1)$ and any real number $r$, we denote by $Q(r) = \mathbb{P}[\xi > r]$, the Gaussian tail probability. Finally we use the notation $\mathbf{A} \preceq \mathbf{B}$ to indicate the loewner order: $\mathbf{B} - \mathbf{A}$ is positive semidefinite.

## 2 Existing work

We focus this discussion on the case of Gaussian design $X$. In this case, the exchangeability condition (2) implies that $[X, \tilde{X}]$ has a covariance matrix of the form

$$\underline{\Sigma} = \begin{bmatrix} \Sigma & \Sigma - \mathrm{diag}(\mathbf{s}) \\ \Sigma - \mathrm{diag}(\mathbf{s}) & \Sigma \end{bmatrix}. \tag{4}$$

As observed in [BC15], positive semi-definiteness of this matrix is equivalent to

$$0 \preceq \mathrm{diag}(\mathbf{s}) \preceq 2\Sigma \tag{5}$$

For some $\mathbf{s} \in \mathbb{R}^p$. As a result, finding a knockoff mechanism consists in finding $\mathbf{s}$.

The seminal work [BC15][CFJL18] introduce the following knockoff mechanisms:

EQUI-KNOCKOFFS: The vector $\mathbf{s}$ is chosen of the form $\mathbf{s} = s\mathbf{1}$ for some $s \geq 0$. In light of (5) the smallest value possible for $s$ is $2\lambda_{\min}(\Sigma)$. Assuming the normalization $\mathrm{diag}(\Sigma) = \mathbf{1}$, [CFJL18] recommend choosing

$$s = 2\lambda_{\min}(\Sigma) \wedge 1, \tag{6}$$

with the goal of minimizing the correlation between $X_j$ and $\tilde{X}_j$.

SDP-KNOCKOFFS: The vector $\mathbf{s}$ is chosen to solve the following semidefinite program:

$$\min \|\mathrm{diag}(\Sigma) - \mathbf{s}\|_1 \quad \text{s.t.} \quad \begin{aligned} 0 &\preceq \mathrm{diag}(\mathbf{s}) \preceq \mathrm{diag}(\Sigma) \\ \mathrm{diag}(\mathbf{s}) &\preceq 2\Sigma. \end{aligned}$$

ASDP-KNOCKOFFS: Assume the normalization $\mathrm{diag}(\Sigma) = \mathbf{1}$. Choose an approximation $\Sigma_{\mathsf{a}}$ of $\Sigma$ (see [CFJL18]) and solve:

$$\begin{aligned} &\text{minimize } \|\mathbf{1} - \hat{\mathbf{s}}\|_1 \\ &\text{subject to } \hat{\mathbf{s}} \geq \mathbf{0}, \mathrm{diag}(\hat{\mathbf{s}}) \preceq 2\Sigma_{\mathsf{a}} \end{aligned}$$

and then solve:

$$\begin{aligned} &\text{minimize } \gamma \\ &\text{subject to } \mathrm{diag}(\gamma\hat{\mathbf{s}}) \preceq 2\Sigma \end{aligned}$$

and put $\mathbf{s} = \gamma\hat{\mathbf{s}}$.

We do not discuss other knockoff constructions, such as the exact construction [CFJL18, Section 3.4.1] and deep knockoff [RSC18], which mostly target at general non-Gaussian distributions.

As alluded, previously, [WBC17] performed power analysis in the linear (fixed $n/p$) regime for $\Sigma = \mathbf{I}_p$, in which case all the above knockoff mechanisms give the same answer of $\mathbf{s} = \mathbf{1}$. For a general $\Sigma$, [FDLL19] derived lower bounds on the power in terms of the minimum eigenvalue of the extended covariance matrix $\underline{\Sigma}$ (no specific knockoff mechanism is assumed).

## 3 Overview of the main results

In the paper, we focus on the so-called linear regime where the sampling $n/p$ converges to a constant $\delta$. We allow for general $\Sigma$ and for simplicity, rather than using the Lasso estimator $\hat{\theta}$ defined in (3), we employ a debiased version [ZZ14, vdGBRD14, JM14]

$$\hat{\theta}^u := \hat{\theta} + \frac{\mathsf{d}}{n}\Sigma^{-1}\mathbf{X}^\top(\mathbf{Y} - \mathbf{X}\hat{\theta}), \tag{7}$$

where $1/\mathsf{d} = 1 - \|\hat{\theta}\|_0/n$. To allow for asymptotic results, we consider a sequence $\{(\Sigma^{(p)}, \theta^{(p)})\}_{p \geq 1}$ where $\Sigma^{(p)}$ are covariance matrices of size $m^{(p)} \times m^{(p)}$ and $\theta^{(p)} \in \mathbb{R}^{(p)}$ are vectors of coefficients. Note that we will only consider the cases where $m^{(p)} = p$ or $m^{(p)} = 2p$, depending on whether we consider predictors with or without knockoffs.

At first glance, it is unclear that for such general sequences, any meaningful result can be said about the debiased Lasso estimator $\hat{\theta}^u$ defined in (7). To overcome this obvious limitation, we consider the asymptotic setting where a *standard distributional limit* exists in the sense [JM14, Definition 4.1].

**Definition 1** (Standard distributional limit). Assume constant sampling rate $n^{(p)} = \delta m^{(p)}$. A sequence $\{(\boldsymbol{\Sigma}^{(p)}, \theta^{(p)})\}_{p \geq 1}$ is said to have a *standard distributional limit* with sparsity $(\alpha, \beta)$, if
(i) there exist $\tau \neq 0$ deterministic and d, possibly random, such that the empirical measure

$$\frac{1}{m^{(p)}} \sum_{j=1}^{m^{(p)}} \delta_{\left(\theta_j, \frac{\hat{\theta}_j^u - \theta_j}{\tau}, (\boldsymbol{\Sigma}^{-1})_{jj}\right)^{(p)}}$$

converges almost surely weakly to a probability measure $\nu$ on $\mathbb{R}^3$ as $p \to \infty$. Here, $\nu$ is the probability distribution of $(\Theta, \Upsilon^{1/2} Z, \Upsilon)$, where $Z \sim \mathcal{N}(0, 1)$, and $\Theta$ and $\Upsilon$ are some random variables independent of $Z$. Moreover, we ask that
(ii) as $p \to \infty$, it holds almost surely that

$$\frac{1}{p} \|\theta^{(p)}\|_0 \to \alpha := \mathbb{P}[|\Theta| > 0], \qquad \text{and} \qquad \frac{1}{p} \|\theta^{(p)}\|_1 \to \beta := \mathbb{E}[|\Theta|].$$

Note that (i) implies that $\liminf_{p \to \infty} \|\theta^{(p)}\|_1 / p \geq \mathbb{E}[|\Theta|]$, and $\liminf_{p \to \infty} \|\theta^{(p)}\|_0 / p \geq \mathbb{P}[|\Theta| > 0]$, almost surely. We further impose that equalities are achieved in (ii).

As mentioned in [JM14], characterizing instances having a standard distributional limit is highly nontrivial. Yet, at least, the definition is non-empty since it contains the case of standard Gaussian design. Moreover, a non-rigorous replica argument indicates that the standard distributional limit exists as long as a certain functional defined on $\mathbb{R}^2$ has a differentiable limit [JM14, Replica Method Claim 4.6], which is always satisfied for block diagonal $\boldsymbol{\Sigma}$ where the empirical distribution of the blocks converges.

We remark that in the sparse regime where $\|\theta\|_0 = o(p)$, rigorous results, that do not appeal to the replica method, show that the weak convergence of the distribution of $\{(\underline{\theta}_j, \mathbf{P}_{jj})\}_{j=1}^p$ is essentially sufficient for the existence of a standard distributional limit ([JM14, Theorem 4.5]), although the present paper does not concern that regime.

We now introduce the key criterion to characterize consistency of a knockoff mechanism and more generally of a variable selection procedure.

**Definition 2** (Effective signal deficiency). For a given variable selection procedure, $\mathsf{ESD}^{(p)} \geq 0$ is a function of $\boldsymbol{\Sigma}^{(p)}$ with the following property: for the class of sequences $(\theta^{(p)}, \boldsymbol{\Sigma}^{(p)})_{p \geq 1}$ satisfying suitable distributional limit conditions, vanishing ESD is equivalent to consistency of the test:

$$\mathsf{ESD} := \limsup_{p \to \infty} \mathsf{ESD}^{(p)} \to 0 \iff \limsup_{p \to \infty} \left\{ \mathsf{FDR}^{(p)} + (1 - \mathsf{PWR}^{(p)}) \right\} \to 0.$$

When we consider knockoff filters, ESD is frequently expressed in terms of the extended covariance matrix $\underline{\boldsymbol{\Sigma}}$, which is in turn a function of $\boldsymbol{\Sigma}$ for a given knockoff mechanism. In that setting, the "suitable distributional limit conditions" in the above definition requires that the sequence of extended instances $(\underline{\theta}^{(p)}, \underline{\boldsymbol{\Sigma}}^{(p)})_{p \geq 1}$ has a standard distributional limit.

Note that by definition, ESD is not unique, and our goal is to find simple representations of its equivalence class. ESD is a potentially useful concept in comparing or evaluating different ways of generating knockoff matrices. As an analogy, think of the various notions of convergences of probability measures. A sequence of probability measures may converge in one topology but not in another. Similarly, one may cook up different functionals of the covariance matrix, such as $\lim_{p \to \infty} p \operatorname{Tr}^{-1}(\boldsymbol{\Sigma})$ and $\lim_{p \to \infty} p \operatorname{Tr}(\boldsymbol{\Sigma}^{-1})$, which both intuitively characterize some sort of signal deficiency since they tend to be small when the signal gets stronger. However, they are not equivalent, and the second convergence to 0 is *stronger* in the sense that the first must vanish when the second vanishes. ESD is intended to be the correct notion of "convergence" that characterizes FDR tending to 0 and power tending to 1.

Of course, by definition it is not obvious that a succinct expression of such an effective signal deficiency exists. Remarkably, we find that the effective signal deficiency can be characterized by the convergence of certain empirical distribution derived from $\boldsymbol{\Sigma}$. The effective signal deficiency for various (old and new) variable selection procedures is as follows:

LASSO: The debiased Lasso [JM14] is a popular method for high-dimensional statistical inference. It is implemented by first computing a Lasso estimator

$$\hat{\theta} = \underset{t \in \mathbb{R}^p}{\operatorname{argmin}} \left\{ \frac{1}{2n} \|\mathbf{Y} - \mathbf{X}\theta\|^2 + \lambda \|\theta\|_1 \right\}$$

where $\lambda > 0$ can be chosen as any fixed positive number independent of $p$. Instead of a direct threshold test on $\hat{\theta}$, we first compute an "unbiased version" $\hat{\theta}^u$ defined in (7), as in [JM14], and pass a threshold to select non-nulls. We show in Theorem 3 and Proposition 4 that we may chose

$$\mathsf{ESD} = \lim_{p \to \infty} d_{\mathsf{LP}} \left( \frac{1}{p} \sum_{j=1}^p \delta_{\mathbf{P}_{jj}^{(p)}}, \delta_0 \right),$$

where $d_{\mathsf{LP}}$ denotes the Lévy-Prokhorov distance between defined for any two measures $\mu$ and $\nu$ defined over a metric space as

$$d_{\mathsf{LP}}(\mu, \nu) := \inf \{ \epsilon > 0 \, : \, \mu(A) \leq \nu(A^\epsilon) + \epsilon, \, \nu(A) \leq \mu(A^\epsilon) + \epsilon, \forall A \},$$

where $A^\epsilon$ denotes the $\epsilon$-neighborhood of $A$. In particular, we have

$$d_{\mathsf{LP}} \left( \frac{1}{p} \sum_{j=1}^p \delta_{\mathbf{P}_{jj}^{(p)}}, \delta_0 \right) := \inf \left\{ \epsilon > 0 \, : \, \frac{\#\{ j \, : \, \mathbf{P}_{jj}^{(p)} \geq \epsilon \}}{p} \leq \epsilon \right\}. \tag{8}$$

The assumption of the standard distributional limit ensures the weak convergence of the empirical distribution of $(\mathbf{P}_{jj}^{(p)})_{j=1}^p$, and hence the convergence of (8). Hereafter, for any vector $x \in \mathbb{R}^m$, we use the shorthand (abusive) notation

$$\|(x_j)_j\|_{\mathsf{LP}} := d_{\mathsf{LP}} \left( \frac{1}{m} \sum_{j=1}^m \delta_{x_j}, \delta_0 \right).$$

This characterization if ESD is, in fact tight: $\mathsf{ESD} \to 0$ is a necessary and sufficient condition for consistency of thresholded Lasso as a variable selection procedure (see Proposition 4)

GENERAL KNOCKOFF: for a general knockoff construction, including variational formulations such as SDP-knockoffs, it seems hopeless to find simple expressions of ESD in terms of $\mathbf{\Sigma}$. Nevertheless, if $(\underline{\theta}^{(p)}, \underline{\mathbf{\Sigma}}^{(p)})$ has a standard distributional limit, we can choose $\mathsf{ESD} = \lim_{p \to \infty} \|(\underline{\mathbf{P}}_{jj}^{(p)})_j\|_{\mathsf{LP}}$ where we recall that $\underline{\mathbf{P}}$ is the extended precision matrix of $[X, \tilde{X}]$.

EQUI-KNOCKOFF: Specializing the above result to the equi-knockoff case, we see that we can choose $\mathsf{ESD} = \lim_{p \to \infty} \lambda_{\max}(\mathbf{P}^{(p)})$, achieved when $s = a\lambda_{\min}(\mathbf{\Sigma})$ for any $a \in (0, 2)$. Note that this is slightly different from the choice (6) prescribed in [BC15, CFJL18] where $s := \min\{1, 2\lambda_{\min}(\mathbf{\Sigma})\}$.

CI-KNOCKOFF: We introduce a new method for generating the knockoff matrix, called *conditional independence knockoff* or CI-knockoff in short. If the Gaussian graphical model associated to $X$ is a tree, i.e. if the sparsity pattern of $\Sigma^{-1}$ corresponds to the adjacency matrix of a tree, then the conditional independence knockoff always exists and $\mathsf{ESD} = \lim_{p \to \infty} \|(\underline{\mathbf{P}}_{jj}^{(p)} \mathbf{\Sigma}_{jj})_j\|_{\mathsf{LP}}$. For example, in the independent case where $\mathbf{\Sigma}$ is diagonal, we get $\mathsf{ESD} = 1$ which readily yields consistency.

The last knockoff construction, conditional independence knockoff, appears to be new. It is both analytically simple and empirically competitive. Comparing equi- and CI- knockoffs: the latter is more robust, since having a small fraction of $j$ with large $\mathbf{P}_{jj}^2 \mathbf{\Sigma}_{jj}$ does not increase its ESD much. For example, two predictors are identical, then the ESD for conditional independence knockoff almost does not change, but equi-knockoff completely fails. Compared to other previous knockoffs, we find that CI-knockoff usually shows similar or improved performance empirically, while being easier to compute and to manipulate.

## 4 Baseline: Lasso with oracle threshold

Consider a variable selection algorithm in which the Lasso parameters with absolute values above a threshold are selected, and suppose that the threshold which controls the FDR is given by an oracle. Note that the knockoff filter is based on the Lasso estimator but it must choose threshold in a data driven fashion. As a result, the Lasso with oracle threshold presents a strong baseline against which the performance of a given knockoff filter should be compared. Not surprisingly, and also as noted in [FDLL19], although the knockoff filter has the advantage of controlling FDR, it usually has a lower power than Lasso with oracle threshold. This fact will become more transparent as we determine their ESD.

**Theorem 3.** *Let $\lambda > 0$ be arbitrary and let $\{(\boldsymbol{\Sigma}^{(p)}, \theta^{(p)})\}_{p \geq 1}$ admit a standard distributional limit, and denote the distributional limit by $(\Theta, \Upsilon^{1/2}Z, \Upsilon)$, where $Z \sim \mathcal{N}(0,1)$, and $\Theta$ and $\Upsilon$ are some random variables independent of $Z$. Assume further that $L := \lim_{p \to \infty} \|(\mathbf{P}_{jj}^{(p)})_j\|_{\mathsf{LP}}$ where the limit exists almost surely by the standard distributional limit assumption. Consider the algorithm which selects $j$ for which $|\hat{\theta}_j^u| \geq t$, where $\hat{\theta}^u$ is defined in (7). Then with the choice of $t = L^{1/4}$,*

$$\limsup_{p \to \infty}\{\mathsf{FDR}^{(p)} + (1 - \mathsf{PWR}^{(p)})\} \leq C_{L,\mu_\Theta,\tau}$$

*where $\lim_{L \to 0} C_{L,\mu_\Theta,\tau} = 0$ for any $\mu_\Theta$ with $\mathbb{P}[|\Theta| > 0] > 0$ and $\tau$ as in the definition of the standard distributional limit. In particular, if $\delta > 1$, then $\tau$ can be bounded in terms of $\sigma$, $\lambda$, $\delta$ and $\mu_\Theta$ only (independent of $\mu_\Upsilon$), and hence $C_{L,\mu_\Theta,\tau}$ in the above inequality can be replaced by $C_{L,\mu_\Theta,\sigma,\lambda,\delta}$ where $\lim_{L \to 0} C_{L,\mu_\Theta,\sigma,\lambda,\delta} = 0$.*

The above theorem implies that $L \to 0$ is a sufficient condition for consistency; this is in fact also necessary, as indicated by the following complementary lower bound.

**Proposition 4.** *(Lower bound). In the previous theorem, assume further that $\Upsilon$ is independent of $\Theta$. Then for any $t > 0$,*

$$\liminf_{p \to \infty}\{\mathsf{FDR}^{(p)} + (1 - \mathsf{PWR}^{(p)})\} \geq c_{L,\sigma,\mu_\Theta}.$$

*where $c_{L,\sigma,\mu_\Theta}$ is increasing in $L$, strictly positive as long as $L > 0$.*

Combining the above two results, we get the following interpretation. Suppose that the distribution of $\Theta$ and the values of $\sigma$ are fixed, and suppose that the parameters $\lambda$ and $t$ in the algorithm optimally tuned (i.e. minimizing $\limsup_{p \to \infty}\{\mathsf{FDR}^{(p)} + (1 - \mathsf{PWR}^{(p)})\}$ for any given distributions). If $\delta > 1$, then, remarkably, the variable selection procedure is consistent if and only if $L$ being small – as long as $\Upsilon$ is independent of $\Theta$, while other characteristics of the law of $\Upsilon$ are not necessary to know. In other words, we proved that $\mathsf{ESD} = L := \lim_{p \to \infty} \|(\mathbf{P}_{jj}^{(p)})_j\|_{\mathsf{LP}}$. If $\delta \leq 1$, small $L$ may not be sufficient for consistency since $C_{L,\mu_\Theta,\sigma,\lambda,\delta}$ also depends on $\mu_\Upsilon$ through $\tau$.

## 5 Results for general knockoff mechanisms

Given $\boldsymbol{\Sigma}$, let $\underline{\boldsymbol{\Sigma}}$ be the extended $2p \times 2p$ covariance matrix for the true predictors and their knockoffs. Let $\underline{\theta} = [\theta, \mathbf{0}] \in \mathbb{R}^{2p}$. Consider the procedure of the knockoff filter described in Section 2, with a slight tweak: define $\Delta_j := |\hat{\underline{\theta}}_j^u| - |\hat{\underline{\theta}}_{j+p}^u|$, where

$$\hat{\underline{\theta}}^u = \hat{\underline{\theta}} + \frac{\mathsf{d}}{n}\underline{\boldsymbol{\Sigma}}^{-1}[\mathbf{X}, \tilde{\mathbf{X}}]^\top(\mathbf{Y} - [\mathbf{X}, \tilde{\mathbf{X}}]\hat{\underline{\theta}})$$

and $\hat{\underline{\theta}}$ is defined in (3). This modification still fulfills the sufficiency and antisymmetry condition in [BC15, Section 2.2], so its FDR can still be controlled. This change allows us to perform analysis using results in [JM14]. We also assume that the Lasso parameter $\lambda$ is an arbitrary number independent of $p$.

**Theorem 5.** *Let $\{(\underline{\boldsymbol{\Sigma}}^{(p)}, \underline{\theta}^{(p)})\}_{p \geq 1}$ admit a standard distributional limit for a given $\lambda \geq 0$, and denote the distributional limit by $(\underline{\Theta}, \underline{\Upsilon}^{1/2}Z, \underline{\Upsilon})$, where $Z \sim \mathcal{N}(0,1)$, and $\underline{\Theta}$ and $\underline{\Upsilon}$ are some random variables independent of $Z$. Assume further that $L := \lim_{p \to \infty} \|(\underline{\mathbf{P}}_{jj}^{(p)})_j\|_{\mathsf{LP}}$ where the limit exists*

*almost surely under the standard distributional limit assumption. Then the knockoff filter with FDR budget $q \in (0, 1)$ satisfies:*

$$\liminf_{p \to \infty} \mathsf{PWR}^{(p)} \geq 1 - C_{L,q,\tau,\mu_{\underline{\Theta}}},$$

*where $\lim_{L \to 0} C_{L,q,\tau,\mu_{\underline{\Theta}}} = 0$ for any given $q$, $\tau$, $\mu_{\underline{\Theta}}$. Further if $\delta > 2$, then $C_{L,q,\tau,\mu_{\underline{\Theta}}}$ in the above inequality can be replaced by $C_{L,q,\lambda,\sigma,\delta,\mu_{\underline{\Theta}}}$.*

Taking $q \to 0$ in the above theorem implies that $L \to 0$ is sufficient for consistency; the following result shows the necessity in a representative setting:

**Proposition 6.** *In the previous theorem, further assume that $\theta_j = \mathbb{1}\{j \in \mathcal{H}_1\}$ where $|\mathcal{H}_1| = \alpha p$ ($\alpha > 0$) is selected uniformly at random. Then, under a suitable distributional limit assumption, the knockoff filter with FDR budget $q \in (0, \alpha L Q^2(\frac{1}{\sigma \sqrt{L}}))$ satisfies:*

$$\limsup_{p \to \infty} \mathsf{PWR}^{(p)} \leq 3/4.$$

The "suitable distributional limit assumption" in Proposition 6 postulates a Gaussian limit for the empirical distribution of the pair $(\hat{\theta}_j^u - \underline{\theta}_j, \hat{\theta}_{j+p}^u - \underline{\theta}_{j+p})_{j=1}^p$, which is stronger than the marginal Gaussian limit assumption in Definition 1, but nevertheless supported by the replica heuristics. Moreover, this condition can be rigorously shown for the case of $\delta > 2$, $\lambda = 0$ (least squares) and block diagonal $\underline{\Sigma}$. The assumption that $\theta_j = 1$ under $\mathcal{H}_1$ in Proposition 6 facilitates the proof but we expect that a similar inconsistency result holds for general $\mu_{\Theta}$. The assumption that $\mathcal{H}_1$ is selected uniformly at random is a counterpart of the independence of $\Theta$ and $\Upsilon$ in Proposition 4.

Together, Theorem 5 and Proposition 6 show that for the knockoff filer, $\mathsf{ESD} = \lim_{p \to \infty} \|(\mathbf{P}_{jj}^{(p)})_j\|_{\mathsf{LP}}$ in the regime of $\delta > 1$. This suggests that one should construct the knockoff variables so that the empirical distribution of $(\underline{\mathbf{P}}_{jj})_{j=1}^{2p}$ converges to 0 weakly.

## 6  Conditional independence knockoff and ESD

We introduce the *conditional independence knockoff*, where $X_j$ and $\tilde{X}_j$ are independent conditionally on $X_{\neg j} := \{X_k, k \in [p] \setminus \{j\}\}$, for each $j = 1, \ldots, p$. This condition implies that

$$\mathbb{E}[X_j \tilde{X}_j] = \mathbb{E}\big[\mathbb{E}[X_j \tilde{X}_j | X_{\neg j}]\big] = \mathbb{E}\big[(\mathbb{E}[X_j | X_{\neg j}])^2\big]$$

Therefore recalling that $s_1, \ldots, s_p$ are as defined in (4), we get

$$\begin{aligned}
s_j &= \mathbf{\Sigma}_{jj} - \mathbb{E}[X_j \tilde{X}_j] \\
&= \mathbb{E}\big[\mathbb{E}[X_j^2 | X_{\neg j}]\big] - \mathbb{E}\big[(\mathbb{E}[X_j | X_{\neg j}])^2\big] \\
&= \mathbb{E}[\mathrm{Var}(X_j | X_{\neg j})] = P_{jj}^{-1}.
\end{aligned} \tag{9}$$

However such an $\mathbf{s}$ may violate the positive semidefinite assumption for the joint covariance matrix (examples exist already in the case $p = 3$). Yet, interestingly, we find that in the case of tree graphical models, this construction always exists. In many practical scenarios, the predictors $X^p$ comes from a tree graphical model, and we can estimate the underlying graph sing the Chow-Liu algorithm [CL68].

**Theorem 7.** *The covariance matrix $\underline{\mathbf{\Sigma}}$ defined in (4) is positive semidefinite with $\mathbf{s}$ defined in (9), if either 1) $\mathbf{\Sigma}$ is the covariance matrix of a tree graphical model; or 2) $\mathbf{P}$ is diagonally dominant.*

Either condition in the theorem intuitive imposes that the graph is sparse. In practice, $\mathbf{\Sigma}$ needs to be estimated, which is generally only feasible with some sparse structure (e.g. via graphical lasso).

Assuming the existence of a standard distributional limit and $\delta > 1$, we have the following results:

**Theorem 8.** *For tree graphical models, $\mathsf{ESD} = \lim_{p \to \infty} \|(\underline{\mathbf{P}}_{jj}^{(p)} \mathbf{\Sigma}_{jj})_j\|_{\mathsf{LP}}$ for CI-KNOCKOFF.*

**Theorem 9.** $\mathsf{ESD} = \lambda_{\max}(\mathbf{\Sigma})$ *for EQUI-KNOCKOFF if $s_j = a\lambda_{\min}(\mathbf{\Sigma})$, $a \in (0, 2)$, $j = 1, \ldots, p$.*

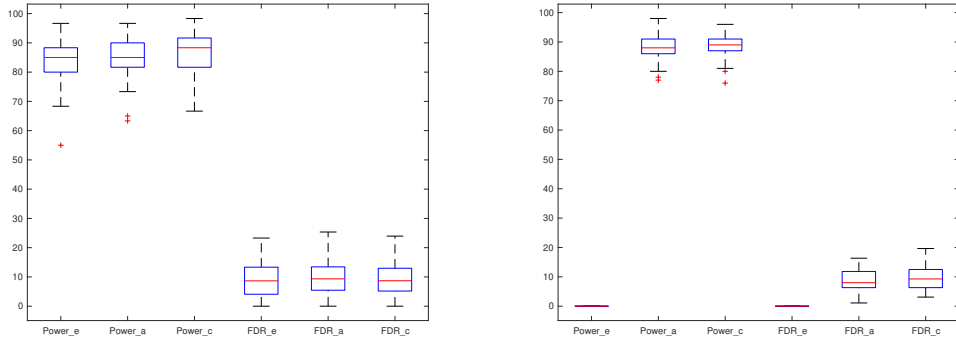

Figure 1: Comparisons of EQUI-KNOCKOFF, ASDP-KNOCKOFF, and CI-KNOCKOFF. Left: Binary tree, equal correlations. Right: Markov chain, randomly chosen correlation strengths.

## 7 Experimental results

First consider the setting where $X_1, \ldots, X_p \sim \mathcal{N}(0, 1)$ and the conditional independence graph forms a binary tree. The correlations between adjacent nodes are all equal to $0.5$. Choose $k = 100$ out of $p = 1000$ indices uniformly at random as the support of $\theta$, and set $\theta_j = 4.5$ for $j$ in the support. Generate $n = 1000$ independent copies of $(X, Y)$ in $Y = X^\top \theta + \xi$ where $\xi \sim \mathcal{N}(0, n)$.

Figure 1, left shows the box plots of the power and FDR for EQUI-KNOCKOFF, ASDP-KNOCKOFF, and CI-KNOCKOFF, where $s$ is defined as in (6) for CI-KNOCKOFF. The FDR is controlled at the target $q = 0.1$ in all three cases. The powers are not statistically significantly different, but the rough trend is $\mathsf{PWR_e} < \mathsf{PWR_a} < \mathsf{PWR_c}$. We then compare the effective signal deficiency. Note that in the current setting, $\mathrm{Var}(\underline{X}_j|\underline{X}_{\neg j}) \leq 1$, and hence $\underline{\mathbf{P}}_{jj} \geq 1$, for each $j = 1, \ldots, 2p$, and we always have $\|(\underline{\mathbf{P}}_{jj})_{j=1}^{2p}\|_{\mathsf{LP}} = 1$ by definition (8), which cannot reveal any useful information for comparison. To resolve this, we can scale down $\underline{\mathbf{P}}_{jj}$ by a common factor before computing the LP distances, noting that it yields a valid effective signal deficiency. Lacking a systematic way of choosing such a scaling factor, heuristically we choose it as 2000 so that the LP distances for the three algorithms are all "bounded away from 0 and 1". We find that $d_{\mathsf{LP,e}} \simeq 0.501$, $d_{\mathsf{LP,a}} \simeq 0.048$ and $d_{\mathsf{LP,c}} \simeq 0.002$ and their ordering matches the ordering of the powers.

In the previous example, the simplest EQUI-KNOCKOFF has a highly competitive performance. However, this is an artifact of the fact that the data covariance is highly structured (i.e., correlations are all the same). If the correlations have high fluctuations, and in particular, a small number of node pairs are highly correlated, then the equi-knockoff has a much worse performance. This is demonstrated in the next example. Consider the setting where $X_1, \ldots, X_p$ forms a Markov chain, in which $X_1, \ldots, X_p \sim \mathcal{N}(0, 1)$. In other words, the Gaussian graphical model is a path graph. The correlation between $X_j$ and $X_{j+1}$ is $\rho_j := G_j \mathbb{1}\{|G_j| \leq 1\}$, where $G_j \sim \mathcal{N}(0, 0.25)$, $j = 1, \ldots, p-1$ are chosen independently. Choose $k = 100$ out of $p = 1000$ indices uniformly at random as the support of $\theta$, and set $\theta_j = 4.5$ for $j$ in the support. Generate $n = 1200$ independent copies of $(X, Y)$ in $Y = X^\top \theta + \xi$ where $\xi \sim \mathcal{N}(0, 0.49n)$.

Figure 1 Right shows the box plots of the power and FDR for the knockoff filter with three different knockoff constructions. The target FDR is $q = 0.1$. Since the correlations are now chosen randomly, with high probability there exist highly correlated nodes, and hence $\lambda_{\min}(\mathbf{\Sigma})$ can be very small, in which case the equi-knockoff performs poorly. However $\mathsf{PWR_c}$ is similar to $\mathsf{PWR_a}$, with the median of the former slightly higher. To compare the ESD, first scale down $\underline{\mathbf{P}}_{jj}$ by a heuristically chosen factor 100. We find $d_{\mathsf{LP,e}} \simeq 0.9995$, $d_{\mathsf{LP,a}} \simeq 0.8660$, and $d_{\mathsf{LP,c}} \simeq 0.1075$ and their ordering matches the ordering of the powers of the three knockoff constructions.

### Acknowledgments

JL was supported by the IDSS Wiener Fellowship. PR was supported by NSF awards IIS-BIGDATA-1838071, DMS-1712596 and CCF-TRIPODS- 1740751; ONR grant N00014-17-1-2147.

## Footnotes

[1] Regression problems with knockoffs are $2p$ dimensional rather than $p$ dimensional. To keep track of this fact, we use $\underline{\cdot}$ to denote a $2p$ dimensional vector.

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
