[Supplementary Material]

# A Perspective on False Discovery Rate Control via Knockoffs

## Abstract

The knockoff filter introduced by Barber and Candès 2016 is an elegant framework for controlling the false discovery rate in variable selection. Yet, there is no conclusive result on the power (type II error rate) analysis, or how to choose the knockoff generation method, even in the Gaussian setting. When the predictors are i.i.d. Gaussian, it is known that as the signal to noise ratio tend to infinity, the power of the knockoff filter tends to 1 under any fixed FDR budget. However, when the predictors have a general covariance structure $\Sigma$, it is not obvious that one can define an analogous notion of the signal to noise ratio. We introduce the notion of *effective signal deficiency* (ESD) as any functional of $\Sigma$, such that the power tend to 1 *if and only if* this functional tends to 0 (under given noise level, sparsity ratio, and sampling rate). We then study the ESD for Lasso and the knockoff filter with different knockoff constructions, assuming the correctness of the replica method prediction for Lasso. As a baseline for comparison, we show that using Lasso with an oracle for choosing the threshold that gives the correct FDR, the ESD tends to 0 if and only if the empirical distribution of the diagonals of the precision matrix $\mathbf{P} := \Sigma^{-1}$ convergences to 0 in distribution. In other words, the ESD can be taken as $\|(P_{jj})_{j=1}^p\|_{LP} := \inf\left\{\epsilon > 0 \colon \frac{1}{p}|\{P_{jj} \geq \epsilon\}| \leq \epsilon\right\}$. For the knockoff filter, if $\underline{\mathbf{P}}$ is the $2p \times 2p$ precision matrix for the predictors and knockoff variables, we show that the ESD is $\|(\underline{P}_{jj})_{j=1}^{2p}\|_{LP}$. We then find more explicit formulae for various specific knockoff constructions. We introduce the *conditional independence knockoff*, which always exists for Gaussian tree graphical models (or when the graph is sufficiently sparse), and show that its ESD is $\|(\Sigma_{jj}P_{jj}^2)_{j=1}^p\|_{LP}$. In contrast, for the equi-knockoffs in the literature, the ESD can achieve $\lambda_{max}(\mathbf{P})$, which is prohibitive when a small set of predictors are highly correlated.

## 1 Introduction

Modern large-scale data analysis often concerns the problem of finding a small set of highly informative predictors, among a larger set often of size comparable or larger than the number of observations. Examples include selecting a few genes related to a certain disease, or discovering a number of demographic attributes linked to the crime rates in a community. False-discovery rate (FDR) control, popularized by Benjamini and Hochberge's [BH95], has become a now-standard criterion for the type I errors in such large-scale hypothesis testing problems. Under orthogonal designs and assuming that the $p$-values under the null hypothesis are known, the Benjamini-Hochberge method is guaranteed to bound the FDR below any desired threshold ([BH95][STS04]). Recently, the knockoff filter [BC15][CFJL18] has emerged as a competitive approach for FDR control, which extends to setting beyond orthogonal designs and known $p$-values under null, and has demonstrated great empirical success [KS][SKB$^+$].

The knockoff filter is build on the Lasso estimator. While it is possible to perform variable selection by a simple threshold test for the Lasso coefficients, it may not be feasible to determine such a threshold that gives rise to the desired FDR. The idea of the knockoff filter is to generate knockoff (fake) variables that has the same distribution as the true ones, but conditionally independent of the observations, and then regress the observations on both the true variables and the knockoff variables. Roughly speaking, one can then determine a threshold with the desired FDR guarantee, by leveraging on the Lasso coefficients for the knockoff variables, knowing that they are nulls.

It is known that any construction of the knockoffs that satisfies a certain exchangeability condition is guaranteed to control the FDR [BC15, Theorem 1]. Such a construction is not unique, and we would like to choose one with a high power (type II error rate). Heuristically, we would like the knockoffs to be as uncorrelated with the true predictors as possible, while the exchangeability condition is satisfied. Various algorithms for generating the knockoffs have been proposed in different settings, which typically involves solving an optimization that minimizes a heuristically chosen correlation measure [BC15][CFJL18][RSC]. Knockoff constructions with analytic expressions are rare (with the exception of the equi-knockoff [BC15] and metropolized knockoff sampling [BCJW19]). Partly due to this, analytical studies of the power of the knockoff filter has been very limited, even in the Gaussian setting. In the special case where the predictors are independent, one can generate the knockoffs simply independent of the true predictors, in which case [WBC17] has shown that the power tends to 1 as as the signal to noise ratio tends to infinity (under a fixed sampling rate), by leveraging results on the Lasso statistics [BM12][SBC17]. For the case of correlated predictors, [FDLL19] proved a lower bound on the power, where the limiting (sample size $n \to \infty$) power is bounded below in terms of the number of predictors $p$ and extremal eigenvalues of the covariance matrix of the true and knockoff variables. The assumption of bounded eigenvalues may not appear to capture the crux of the matter in certain scenarios. For example, if all predictors are independent except that two of them are always equal, then the minimum eigenvalue of the covariance matrix is zero, yet the FDR and the power is almost unchanged as we experimentally observed for the knockoff filter.

In this paper, we again consider the knockoff filter in the Gaussian case, but ask the following question: for a fixed sampling rate $n/p$ and given a sequence of predictor covariance matrices $(\mathbf{\Sigma}^{(p)})_{p=1}^{\infty}$, is power of the knockoff filter tending to 1 under any fixed FDR budget? Using the results on the Lasso statistics in [JM14], we find that the answer essentially depends on whether the empirical distribution of the diagonals of $\underline{\mathbf{P}} := \underline{\mathbf{\Sigma}}^{-1}$ converges to 0, where $\underline{\mathbf{\Sigma}}$ denotes the covariance matrix of the true and knockoff variables. Note that $\underline{\mathbf{\Sigma}}^{-1}$ depends on the method of generating knockoffs, and hence this observation can be useful in the comparison of various knockoff constructions; an explicit evaluation function will be provided in (13), which we call *effective signal deficiency*.

A second contribution is to propose a new rule of generating the knockoffs, called *conditionally independent knockoffs* (CIK), which possesses both simple analytic expressions and excellent experimental performance. CIK does not exist for all $\mathbf{\Sigma}$, but we show its existence for tree graphical models or other sufficiently sparse graphs. Note that in practice, the so-called model-X knockoff filter requires the knowledge of $\mathbf{\Sigma}$, an estimation of which is often prohibitive except when the graph has sparse or tree structures. CIK has simple explicit expressions of the effective signal deficiency for tree models, since the empirical distribution of the diagonals of $\mathbf{\Sigma}^{-1}$ is the same as that of $(P_{jj}^2 \Sigma_{jj})_{j=1}^{p}$. We remark that CIK is different than *metropolized knockoff sampling* studied in [BCJW19] (originally appeared in [CFJL18, Section 3.4.1]), even in the case of Gaussian Markov chains. The latter exists for generic distributions and is computationally efficient for Markov chains.

## 2 Preliminaries on the knockoff filter

Notation: $[n] := \{1, \ldots, n\}$. We use boldface such as $\mathbf{X} := (X_{ij})_{i \in [n], j \in [p]}$ and $\mathbf{Y} := (Y_i)_{i \in [n]} = Y^n$ to denote matrix and vectors. $\|\theta\|_0$ and $\|\theta\|_1$ denote the standard $\ell_0$ and $\ell_1$ norms of vectors. $\mathrm{diag}(\mathbf{s})$ is a diagonal matrix when $\mathbf{s}$ is a vector, and $\mathrm{diag}(\mathbf{P})$ denotes the vector of diagonal entries when $\mathbf{P}$ is a matrix. In discussions of knockoffs, we use the underline to indicate instances in the extended cases with knockoff variables, e.g., $\underline{\theta}$ denotes a $2p$-vector when $\theta$ is a $p$-vector. $\mathrm{Q}(r)$, $r \in \mathbb{R}$ denotes the Gaussian tail probability. $\mathbf{A} \preceq \mathbf{B}$ means that the matrix $\mathbf{B} - \mathbf{A}$ is positive semidefinite.

Suppose that the true observation model is $Y = \sum_{j=1}^{p} \theta_{0,j} X_j + N$, where $\theta_0 \in \mathbb{R}^p$ are the unknown parameters, $X^p := (X_j)_{j=1}^{p}$ are the (observable) predictors, and $N$ is the noise. We adopt the model-

X framework [CFJL18], where $X^p$ are assumed to be random variables with known distribution. This is a "semi-supervised learning" setting where a large number of unlabelled samples are available for estimating the distribution of $X^p$. Knowing a sample size $n$ number of observations and predictor values, the knockoff filter aims to determine the set of active predictor, $\{j\colon \theta_{0,j} \neq 0\}$, while controlling the false discovery rate (FDR)

$$FDR := \mathbb{E}\left[\frac{|\mathcal{H}_0 \cap \hat{\mathcal{H}}_1|}{|\hat{\mathcal{H}}_1|}\right] \tag{1}$$

below a given threshold. Here, $H_0 := \{j\colon \theta_{0,j} = 0\}$ and $\hat{\mathcal{H}}_1$ denotes the set of selected predictors. The method is to generate knockoff variables $\tilde{X}_1, \ldots, \tilde{X}_p$, with the property that

$$(X^p, \tilde{X}^p)_{\mathsf{swap}(S)} = (X^p, \tilde{X}^p) \tag{2}$$

in distribution, for any set $S \in \{1, \ldots, p\}$. The swap operation means switching the true and knockoff coordinates with indices in $S$; for example, if $p = 2$ and $S = \{1\}$, then $(X^2, \tilde{X}^2)_{\mathsf{swap}(S)} = (\tilde{X}_1, X_2, X_1, \tilde{X}_2)$.

Recall that we use the underline to indicate instances in the extended cases with knockoff variables. For example, $\underline{\theta}$ is a $2p$-vector, $\underline{\Sigma}$ is a $2p \times 2p$ matrix. The the knockoff filter performs the following: regress $Y$ on $[X^p, \tilde{X}^p]$, let $\underline{\hat{\theta}}_1, \ldots \underline{\hat{\theta}}_{2p}$ be the Lasso coefficients, and put

$$W_j := |\underline{\hat{\theta}}_j| - |\underline{\hat{\theta}}_{j+p}|, \tag{3}$$

$j = 1, \ldots, p$. Choose the data dependent threshold $T > 0$ by the following rule

$$T := \min\left\{t \in \mathcal{W}\colon \frac{|\{j\colon W_j \leq -t\}|}{|\{j\colon W_j \geq t\}| \vee 1} \leq q\right\} \tag{4}$$

where $\mathcal{W} := \{|W_j|\colon j = 1, \ldots, p\} \setminus \{0\}$, and $q$ equals the given FDR budget. Then select $j$ for which $W_j > T$ as the active predictors. It is shown in [BC15, Theorem 1] that this procedure bounds[1] FDR below $q$. More generally, the FDR is controlled below $q$ as long as $(W_j)_{j=1}^p$ depends on $(\mathbf{X}, \mathbf{Y})$ only through $(\mathbf{X}^\top \mathbf{X}, \mathbf{X}^\top \mathbf{Y})$, and satisfies the antisymmetry property [BC15, Section 2.2].

For Gaussian $X^p$, note that the exchangeability condition implies that the covariance of $(X^p, \tilde{X}^p)$ has the form

$$\underline{\Sigma} = \begin{bmatrix} \Sigma & \Sigma - \mathrm{diag}(\mathbf{s}) \\ \Sigma - \mathrm{diag}(\mathbf{s}) & \Sigma \end{bmatrix}. \tag{5}$$

As observed in [BC15], positive semi-definiteness of this matrix is equivalent to

$$\mathrm{diag}(\mathbf{s}) \succeq \mathbf{0}, \tag{6}$$
$$2\Sigma - \mathrm{diag}(\mathbf{s}) \succeq \mathbf{0}. \tag{7}$$

Previous methods for generating the knockoffs (computing $\mathbf{s}$) include the following [CFJL18]:

- The equi-knockoffs construction chooses $s_1, \ldots, s_p$ all equal. Note that the maximum of such value compatible with (7) is $2\lambda_{min}(\Sigma)$. [CFJL18] assumed the normalization $\Sigma_{jj} = 1$, $j = 1, \ldots, p$, and recommended choosing

$$s_j = 2\lambda_{min}(\Sigma) \wedge 1, \tag{8}$$

  with the goal of minimizing the correlation between $X_j$ and $\tilde{X}_j$.

- The semidefinite program (SDP) construction solves the following

$$minimize \quad \sum_{j=1}^p |\Sigma_{jj} - s_j| \tag{9}$$

$$s.t. \quad 0 \leq s_j \leq \Sigma_{jj}, \quad \mathrm{diag}(\mathbf{s}) \preceq 2\Sigma. \tag{10}$$

118  • The approximate semidefinite program (ASDP) construction first solves (9) and (10) with
119    $\boldsymbol{\Sigma}$ replaced by a certain block diagonal matrix, returning a vector $\hat{\mathbf{s}}$. Then the final result is
120    chosen as $\mathbf{s} = \gamma\hat{\mathbf{s}}$, where $\gamma$ is the maximum scalar that fulfils (7).

121  We do not discuss other knockoff constructions, such as the exact construction [CFJL18, Section 3.4.1]
122  and deep knockoff [RSC], are mostly targeting at general non-Gaussian distributions.

123  Define

$$POWER := \mathbb{E}\left[\frac{|\hat{\mathcal{H}}_1 \cap \mathcal{H}_1|}{|\mathcal{H}_1|}\right] \tag{11}$$

124  where $H_1 := \{j : \theta_{0,j} \neq 0\}$. Previously, [JM14] performed power analysis for i.i.d. design in the
125  linear (fixed $n/p$) regime. [FDLL19] performed power analysis for a general $\boldsymbol{\Sigma}$, and showed a
126  consistency result as $n \to \infty$.

## 3  Summary of the main results

128  The present paper is interested in the linear (fixed $n/p$) regime and general $\boldsymbol{\Sigma}$. At first glance, it is
129  not even obvious that any meaningful result can be said for a general sequence $(\boldsymbol{\Sigma}^{(p)})_{p=1}^{\infty}$. A starting
130  point is the observation that, under mild conditions, the empirical distribution of the errors in the
131  regression coefficients divided by $(\boldsymbol{\Sigma}^{-1})_{jj}^{1/2}$ is asymptotically normal. This has been hinted or has
132  explicitly appeared in various literature on regression problems, e.g. [JM14][EKBB$^+$13, Lemma 1].

133  More formally, we consider the asymptotic setting where the sequence of instances $\{(\underline{\theta}_0^{(p)}, \underline{\boldsymbol{\Sigma}}^{(p)})\}_{p\geq 1}$
134  has a *standard distributional limit* in the sense of [JM14, Definition 4.1] (reproduced in Definition 3).
135  This assumption implies that the empirical distribution of $\{(\underline{\theta}_{0,j}, \underline{P}_{jj})\}_{j=1}^{p}$ converges weakly to
136  some probability measure on $\mathbb{R}^2$, and it is believed to be not much stronger than that, although a
137  more precise characterization of such sequences remains an outstanding question [JM14]. Assuming
138  the correctness of the replica method calculations, [JM14, Replica Method Claim 4.6] provided
139  mild conditions under which the standard distributional limit exists. In particular, as observed by
140  [JM14], those conditions are satisfied for block diagonal covariance matrices in which the empirical
141  distribution of the block converges. In other words, in a "direct sum" version of the problem where
142  we have a fixed $\boldsymbol{\Sigma}$ but have $k$ independent copies of those predictors, and let $n$ also grow linearly
143  in $k$, the assumption for the replica method claim in [JM14, Replica Method Claim 4.6] is always
144  satisfied. We remark that in the regime of vanishing $\|\theta_0\|_0/p$, there are also rigorous (without
145  appealing to the replica method) results showing that the weak convergence of the distribution of
146  $\{(\underline{\theta}_{0,j}, \underline{P}_{jj})\}_{j=1}^{p}$ is essentially sufficient for the existence of a standard distributional limit ([JM14,
147  Theorem 4.5]), although the present paper does not concern that regime. We introduce:

148  **Definition 1** (Effective signal deficiency)**.** Given any sampling rate[2] $\delta := n/p > 1$, noise level
149  $N_i \sim \mathcal{N}(0, n\sigma^2)$ (under the model $Y_i = \sum_{j=1}^{p} \theta_{0,j} X_{ij} + N_i, i = 1, \ldots, n$), and a variable selection
150  algorithm, define the *effective signal deficiency* (ESD) as any function of a sequence $(\theta_0^{(p)}, \boldsymbol{\Sigma}^{(p)})_{p\geq 1}$
151  with a standard distributional limit, such that the following property holds: for any $\epsilon > 0$, there exists
152  $\epsilon' > 0$ such that $ESD < \epsilon'$ ensures that $\limsup_{p\to\infty} \max\{FDR^{(p)}, 1 - POWER^{(p)}\} < \epsilon$.

153  We are often interested in settings where $\theta_0$ has given sparsity level and bounds on the amplitude of
154  the nonzero coefficients, so effectively ESD is a function of the sequence $(\boldsymbol{\Sigma}^{(p)})_{p\geq 1}$. Also note that
155  by definition, ESD is not unique, and our goal is to find simple representations of the equivalent class.
156  ESD is a potentially useful concept in comparing or evaluating different ways of generating knockoff
157  matrices. As an analogy, think of the various notions of convergences of probability measures.
158  A sequence of probability measures may converge in one topology but not in another. Similarly,
159  one may cook up different functionals of the covariance matrix, such as $\lim_{p\to\infty} p\,\mathrm{Tr}^{-1}(\boldsymbol{\Sigma})$ and
160  $\lim_{p\to\infty} p\,\mathrm{Tr}(\boldsymbol{\Sigma}^{-1})$, which both intuitively characterize some sort of signal deficiency since they
161  tend to be small when the signal gets stronger. However, they are not equivalent, and the second
162  convergence to 0 is *stronger* in the sense that the first must vanish when the second vanishes. ESD is

163 intended to be the correct notion of "convergence" that characterizes FDR tending to $0$ and power
164 tending to $1$.

165 Of course, by definition it is not obvious that a succinct expression of such an effective signal
166 deficiency exists. Remarkably, we find that the effective signal deficiency can be characterized by
167 the convergence of certain empirical distribution derived from $\mathbf{\Sigma}$. The effective signal deficiency for
168 various (old and new) algorithms as follows:

- 169 Not using knockoffs: one may use Lasso to regress $\mathbf{Y}$ on $\mathbf{X}$, and obtain $\hat{\theta}$:

$$\hat{\theta} = \operatorname{argmin}_{\theta \in \mathbb{R}^p} \left\{ \frac{1}{2n} \|\mathbf{Y} - \mathbf{X}\theta\|^2 + \lambda\|\theta\|_1 \right\} \tag{12}$$

170 The parameter $\lambda$ can be chosen as any fixed positive number independent of $p$. Instead of
171 a direct threshold test on $\hat{\theta}$, we compute an "unbiased version" $\hat{\theta}^u$ (defined in (15)) as in
172 [JM14] for simplicity of the analysis, and pass a threshold to select non-nulls. Suppose that
173 there is an oracle telling how to pick the threshold to make FDR at the desired level. We
174 show that ESD for this oracle algorithm is the limit $p \to \infty$ of

$$\|(P_{jj})_{j=1}^p\|_{LP} := \inf \left\{ \epsilon > 0 : \frac{1}{p}|\{P_{jj} \geq \epsilon\}| \leq \epsilon \right\}. \tag{13}$$

175 The assumption of the standard distributional limit ensures the weak convergence of the
176 empirical distribution of $(P_{jj})_{j=1}^p$, and hence the convergence of (13). For simplicity, we
177 may simply say $\|(P_{jj})_{j=1}^p\|_{LP}$ is ESD without mentioning the limit in $p$, when there is no
178 confusion. In other words, we defined $\|(P_{jj})_{j=1}^p\|_{LP}$ as the distance between the empirical
179 distance of $\operatorname{diag}(\mathbf{P})$ and the delta measure at 0, under the Lévy-Prokhorov metric[3] (we
180 are abusing the notation of norms even though this is not a norm). Note that $\|\|_{LP}$ can be
181 replaced by any metric compatible with the weak convergence topology.

- 182 General knockoff: for a general (potentially non-analytic) knockoff construction, it seems
183 hopeless to find simple expressions of ESD in terms of $\mathbf{\Sigma}$. Nevertheless, if $(\underline{\theta}_0^{(p)}, \underline{\mathbf{\Sigma}}^{(p)})$ has
184 a standard distributional limit, we can express ESD as $\|(\underline{P}_{jj})_{j=1}^{2p}\|_{LP}$, where we recall that
185 $\underline{\mathbf{P}}$ is the extended precision matrix including the knockoff variables. We next find more
186 explicit expressions in terms of $\mathbf{P}$ for specific constructions:

- 187 Equi-knockoff: we show that ESD is at least $\lambda_{max}(\mathbf{P})$. This is also achievable by choosing
188 $s_j = \lambda_{min}(\mathbf{\Sigma})$ (note that this is slightly different than (8) in [BC15][CFJL18]).

- 189 We introduce a new method for generating the knockoff matrix, called *conditional indepen-*
190 *dence knockoff*. If the Gaussian graphical model is from a tree, the conditional independence
191 knockoff always exists, and the ESD is $\|P_{jj}^2 \Sigma_{jj}\|_{LP}$.

192 As noted in [FDLL19], although knockoff filter has the advantage of controlling FDR, it usually has
193 a lower power than Lasso with oracle threshold. We use oracle threshold Lasso as a baseline for
194 comparison, and indeed its ESD is smaller than that of other algorithms.

195 The last knockoff construction, conditional independence knockoff, appears to be new. It is both
196 analytically simple and empirically competitive. Comparing equi- and conditional independence
197 knockoff: the latter is more robust, since having a small fraction of $j$ with large $P_{jj}^2 \Sigma_{jj}$ does not
198 increase $\|\cdot\|_{LP}$ much. For example, if the $p$ and $p-1$ th predictors are equal, then the ESD for
199 conditional independence knockoff almost does not change, but equi-knockoff completely fails.
200 While the solution in the (approximate) semidefinite knockoff is not analytically simple, empirically
201 we find that the conditional independence knockoff usually has similar or improved performance.

## 202 4 Baseline: Lasso with oracle threshold

203 Before analyzing any algorithm, let us observe the following converse bound, which is information-
204 theoretic (i.e. not limited to Lasso or any particular algorithm). This result lower bounds the effective

signal deficiency (ESD) by $\|(P_{jj})_{j=1}^p\|_{LP}$, where $\|\|_{LP}$ was defined in (13). Intuitively, the result comes from the fact that the conditional variance of $X_j$ given $X_{\setminus j}$ is $P_{jj}^{-1}$.

**Proposition 2** (Converse). *Fix $\alpha \in (0,1)$, $n, p \in \mathbb{N}$, and let $\Sigma$ be the covariance matrix of the Gaussian predictors. Let $\theta_1, \ldots, \theta_p$ be i.i.d. $Ber(\alpha)$. Assume that the noise variance (for each sample) is $n$. Assume that there exists an algorithm satisfying $FDR \leq q$, $POWER \geq 1 - \epsilon$. Then for $n, p \geq N(\epsilon, \alpha)$ large enough, we have*

$$\|(P_{jj})_{j=1}^p\|_{LP} \leq \max \left\{ \frac{1.1}{\left(2Q^{-1}(\sqrt{\frac{q}{1-\alpha}} + \sqrt{2\epsilon})\right)^2}, \frac{q}{1-\alpha} + \sqrt{\epsilon} \right\}, \tag{14}$$

*where $Q(a) :=$ denotes the standard Gaussian tail probability. In particular, $\max\{\epsilon, q\} \to 0$ implies that $\|(P_{jj})_{j=1}^p\|_{LP} \to 0$.*

We next show that $\|(P_{jj})_{j=1}^p\|_{LP}$ is also an achievable ESD, and in fact achievable by appropriately using Lasso. More precisely, for a sequence of instances having a standard distributional limit defined as follows (introduced by [JM14, Definition 4.1]), $\lim_{p\to\infty} \|(P_{jj})_{j=1}^p\|_{LP}$ is an achievable ESD, where the existence of the limit is ensured by the standard distributional limit assumption.

**Definition 3** (Standard distributional limit). A sequence $\{(\Sigma^{(p)}, \theta_0^{(p)}, m^{(p)}, n^{(p)}, \sigma^{(p)})\}_{p \geq 1}$ is said to have a *standard distributional limit* if there exists $\tau \neq 0$ and potentially random $\mathsf{d} \in \mathbb{R}$ such that $\{\theta_{0,j}, (\hat{\theta}_j^u - \theta_{0,j})/\tau, (\Sigma^{-1})_{jj}\}_{j=1}^m$ converges almost surely to a probability measure $\nu$ on $\mathbb{R}^3$. Here, $\Sigma^{(p)}$ is $m^{(p)} \times m^{(p)}$ matrix, $\hat{\theta}^u$ is defined in terms of the Lasso estimator:

$$\hat{\theta}^u := \hat{\theta} + \frac{\mathsf{d}}{n} \Sigma^{-1} X^\top (Y - X\hat{\theta}), \tag{15}$$

$\nu$ is the probability distribution of $(\Theta_0, \Upsilon^{1/2} Z, \Upsilon)$, where $Z \sim \mathcal{N}(0,1)$, and $\Theta_0$ and $\Upsilon$ are some random variables independent of $Z$.

As mentioned in [JM14], characterizing instances having a standard distributional limit is highly nontrivial. Yet, at least, the definition is non-empty since it contains the case of standard Gaussian design. Moreover, a non-rigorous replica argument indicates that the standard distributional limit exists as long as a certain functional defined on $\mathbb{R}^2$ has a differentiable limit [JM14, Replica Method Claim 4.6], which is always satisfied for block diagonal $\Sigma$ where the empirical distribution of the blocks converges.

We now consider a simple variable selection algorithm based on the Lasso estimator (without using knockoffs), where indices $j$ for which $|\hat{\theta}_j^u|$ exceeds a certain threshold are selected (see definition of $\hat{\theta}^u$ in (15)). In practice, the knockoff filter has the advantage of controlling FDR. However, if Lasso is used with the right threshold giving the correct FDR, then Lasso has higher power than the knockoff filter (see the discussion in [FDLL19]).

**Proposition 4** (Lasso achievability). *Let $\{(\Sigma^{(p)}, \theta_0^{(p)}, n^{(p)}, \sigma^{(p)})\}_{p \geq 1}$ be any sequence having a standard distributional limit, where*

$$|\theta_{0,j}| \geq 1, \quad \forall j : \theta_{0,j} \neq 0, \tag{16}$$

$$\limsup_{p\to\infty} \|\theta_0^{(p)}\|_1/p = \beta < \infty, \tag{17}$$

$$\lim_{p\to\infty} \|\theta_0^{(p)}\|_0/p = \alpha, \tag{18}$$

$$\lim_{p\to\infty} n^{(p)}/p = \delta, \tag{19}$$

$$\sigma^{(p)} = \sqrt{n}\sigma_0, \tag{20}$$

$$\lim_{p\to\infty} \|(P_{jj})_{j=1}^p\|_{LP} = L. \tag{21}$$

*Then using a threshold test for $\hat{\theta}^u$ defined in (15), where the Lasso parameter $\lambda > 0$ is any number independent of $p$, one can bound both $FDR^{(p)}$ and $1 - POWER^{(p)}$ by $f_{\alpha,\beta,\delta,\sigma_0,\lambda}(L)$ almost surely for large enough $p$, where $f_{\alpha,\beta,\delta,\sigma_0,\lambda}(\cdot)$ is a function that vanishes at the origin for any fixed $\alpha, \beta, \delta, \sigma_0, \lambda$. For explicit bounds, see (22) and (23).*

Explicitly, we have the following bounds almost surely,

$$\limsup_{p \to \infty} FDR^{(p)} \leq 2Q\left(\frac{1}{2\tau\sqrt{L}}\right) + L; \tag{22}$$

$$\liminf_{p \to \infty} POWER^{(p)} \geq 1 - \frac{1}{\alpha}\left[2Q\left(\frac{1}{2\tau\sqrt{L}}\right) + L\right], \tag{23}$$

where $\tau$ is from the definition of the standard distributional limit, and is bounded by

$$\tau^2 \leq \frac{2\delta\sigma_0^2}{\delta - 1} + \frac{2\lambda\delta(\delta + 1)\beta}{(\delta - 1)^3}. \tag{24}$$

## 5 Results for general knockoff matrices

Given $\boldsymbol{\Sigma}$, let $\underline{\boldsymbol{\Sigma}}$ be the extended $2p \times 2p$ covariance matrix for the true predictors and their knockoffs. Let $\underline{\theta}_0$ be the $2p$-vector where the indices corresponding to the knockoff variables ($j = p+1, \ldots, 2p$) are $0$. Consider the procedure of the knockoff filter described in Section 2, with a slight tweak: define $W_j := |\hat{\underline{\theta}}_j^u| - |\hat{\underline{\theta}}_{j+p}^u|$, instead of (3), where unbiased regression coefficients $\hat{\underline{\theta}}^u$ is defined analogous to (15). This definition of $W_j$ still fulfills the sufficiency and antisymmetry condition in [BC15, Section 2.2], so FDR can still be controlled. This change allows us to perform analysis using results in [JM14]. We also assume that the Lasso parameter $\lambda$ is an arbitrary number independent of $p$. Then, assuming the existence of standard distributional limit, we show that $\|(\underline{P}_{jj})_{j=1}^{2p}\|_{LP}$ is ESD.

**Lemma 5.** *Let $\{(\underline{\boldsymbol{\Sigma}}^{(p)}, \theta_0^{(p)}, 2p, n^{(p)}, \sigma^{(p)})\}_{p\geq 1}$ be a sequence having a standard distribution limit. Suppose that* (16)-(20) *still applies, with $\delta > 2$. Suppose that*

$$q > \frac{2Q(1/3\tau\sqrt{L}) + L}{\alpha/2 - 4Q(1/3\tau\sqrt{L}) - 2L} \tag{25}$$

*where $\tau$ is from the definition of the standard distributional limit, which is bounded as in* (24). *Let $L := \lim_{p\to\infty} \|(\underline{P}_{jj})_{j=1}^{2p}\|_{LP}$. Then almost surely, running the knockoff filter with FDR budget at $q$ achieves power*

$$\liminf_{p \to \infty} POWER^{(p)} \geq 1 - \frac{8}{\alpha}Q(1/3\tau\sqrt{L}) - \frac{4L}{\alpha}. \tag{26}$$

*In particular, the asymptotic power tends to $1$ as $L \to 0$.*

## 6 Conditional independence knockoff and ESD

We introduce the *conditional independence knockoff*, where $X_j$ and $\tilde{X}_j$ are conditionally independent of $X_{\backslash j}$, for each $j = 1, \ldots, p$. This condition implies that

$$s_j = \text{Var}(X_j | X_{\backslash j}) = P_{jj}^{-1}, \quad j = 1, \ldots, p, \tag{27}$$

where $s_1, \ldots, s_p$ are as defined in (5). However such an $\mathbf{s}$ may violate the positive semidefinite assumption for the joint covariance matrix (example with $p = 3$ exists). Yet, interestingly, we find that in the case of tree graphical model, this construction always exists. In many practical scenarios, the predictors $X^p$ comes from a tree graphical model, and we can estimate the underlying graph sing the Chow-Liu algorithm [CL68].

**Theorem 6.** $\underline{\boldsymbol{\Sigma}}$ *defined in* (5) *is positive semidefinite with $\mathbf{s}$ defined in* (27)*, if either of the following is satisfied: 1) $\boldsymbol{\Sigma}$ is the covariance matrix of a tree graphical model; 2) $\mathbf{P}$ is diagonally dominant.*

Either condition in the theorem intuitive imposes that the graph is sparse. In practice, $\boldsymbol{\Sigma}$ needs to be estimated, which is generally only feasible with some sparse structure (e.g. via graphical lasso).

Assuming the existence of a standard distributional limit, we have the following results:

**Theorem 7.** *For tree graphical models, assuming that the algorithm is knockoff filter using the conditional independence knockoff, $\|(P_{jj}^2 \Sigma^{jj})_{j=1}^p\|_{LP}$ is an effective signal deficiency.*

**Theorem 8.** *The effective signal deficiency for equi-knockoff with $s_j = \lambda_{min}(\boldsymbol{\Sigma})$, $j = 1, \ldots, p$ is $\lambda_{min}(\boldsymbol{\Sigma})$.*

Figure 1: Left: Binary tree, equal correlations. Extension _e, _a, _c refers to equi-knockoffs, asdp knockoffs, and conditional independence knockoff, respectively. Right: Markov chain, randomly chosen correlation strengths.

## 7  Experimental results

We first consider the setting where the conditional independence graph $X_1, \ldots, X_p$ forms a binary tree, in which $X_1, \ldots, X_p \sim \mathcal{N}(0,1)$. The correlations between adjacent nodes are all equal to $0.5$. Choose $k = 100$ out of $p = 1000$ indices uniformly at random as the support of $\theta$, and set $\theta_j = 4.5$ for $j$ in the support. Generate $n = 1000$ samples $Y_i = \mathbf{X}_i \theta + N_i$ where $N_i \sim \mathcal{N}(0, n)$.

Figure 1 Left shows the box plots of the power and FDR for the knockoff filter with three different knockoff constructions, equi-knockoff, sdp-knockoff, and conditional independence knockoff. The FDR is controlled at the target $q = 0.1$ in all three cases. The powers are similar, but the rough trend is $POWER_{\mathsf{e}} < POWER_{\mathsf{s}} < POWER_{\mathsf{c}}$. We then compare the effective signal deficiency. Note that in the current setting, $\mathrm{Var}(\underline{X}_j | \underline{X}_{\backslash j}) \leq 1$, and hence $\underline{P}_{jj} \geq 1$, for each $j = 1, \ldots, 2p$, and we always have $\|(\underline{P}_{jj})_{j=1}^{2p}\|_{LP} = 1$ by the definition (13), which cannot reveal any useful information for comparison. To resolve this, we can scale down $\underline{P}_{jj}$ by a common factor before computing the $LP$ norms, noting that such a scaled version of the $LP$ norm is still a valid effective signal deficiency (in the same equivalence class). Lacking a systematic way of choosing such a scaling factor, heuristically we choose it so that the LP norms for the three algorithms are all "bounded away from 0 and 1". We find that $\|(\underline{P}_{\mathsf{e},jj})_{j=1}^{2p}/2000\|_{LP} = 0.5010$, $\|(\underline{P}_{\mathsf{s},jj})_{j=1}^{2p}/2000\|_{LP} = 0.0480$, and $\|(\underline{P}_{\mathsf{c},jj})_{j=1}^{2p}/2000\|_{LP} = 0.0025$, and their ordering matches the ordering of the powers.

In the previous example, the simplest equi-knockoff has a highly competitive performance. However, this is an artifact of the fact that the data covariance is highly structured (i.e., correlations are all the same). If the correlations have high fluctuations, and in particular, a small number of node pairs are highly correlated, then the equi-knockoff has a much worse performance. This is demonstrated in the next example. Consider the setting where $X_1, \ldots, X_p$ forms a Markov chain, in which $X_1, \ldots, X_p \sim \mathcal{N}(0,1)$. The correlation between $X_j$ and $X_{j+1}$ is $\rho_j := G_j 1\{|G_j| \leq 1\}$, where $G_j \sim \mathcal{N}(0, 0.25)$, $j = 1, \ldots, p-1$ are chosen independently. Choose $k = 100$ out of $p = 1000$ indices uniformly at random as the support of $\theta$, and set $\theta_j = 4.5$ for $j$ in the support. Generate $n = 1200$ samples $Y_i = \mathbf{X}_i \theta + N_i$ where $N_i \sim \mathcal{N}(0, 0.49n)$.

Figure 1 Right shows the box plots of the power and FDR for the knockoff filter with three different knockoff constructions. The target FDR $q = 0.1$. Since the correlations are now chosen randomly, with high probability there exist highly nodes, and hence $\lambda_{min}(\boldsymbol{\Sigma})$ can be very small, in which case the equi-knockoff performs poorly. (Figure 1 shows the case where the correlations are truncated between $[-1, 1]$. If we truncate the correlation to a smaller interval around $0$, we can observe that $POWER_{\mathsf{e}}$ goes up). However $POWER_{\mathsf{c}}$ is similar to $POWER_{\mathsf{s}}$, with the median of the former slightly higher. To compare the ESD, first scale down $\underline{P}_{jj}$ by a heuristically chosen factor. We find $\|(\underline{P}_{\mathsf{e},jj})_{j=1}^{2p}/100\|_{LP} = 0.9995$, $\|(\underline{P}_{\mathsf{s},jj})_{j=1}^{2p}/100\|_{LP} = 0.8660$, and $\|(\underline{P}_{\mathsf{c},jj})_{j=1}^{2p}/100\|_{LP} = 0.1075$, and their ordering matches the ordering of the powers of the three knockoff constructions.

## Footnotes

[1]While [BC15] discusses fixed knockoff whereas the present paper mainly concerns the model-X knockoff, the proof in [BC15] still works in the model-X case (see the explanation in [CFJL18]).

[2]We assume $\delta > 1$ for convenience so that the parameter $\tau$ in the replica analysis can be bounded independently of $\boldsymbol{\Sigma}$. However the result in [JM14, Definition 4.1] applies to any $\delta > 0$.

[3]Generally, the Lévy-Prokhorov distance between two probability measures $\mu$ and $\nu$ is defined as $\inf\{\epsilon > 0 | \mu(A) \leq \nu(A^\epsilon) + \epsilon, \nu(A) \leq \mu(A^\epsilon) + \epsilon, \forall A\}$, where $A^\epsilon$ denotes the $\epsilon$-neighborhood of $A$.

[4] $o_p(1;\epsilon,p)$ means a sequence indexed by p, which vanishes for any fixed $\epsilon, p$.

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

# A   Proof of Proposition 2

*Proof.* We can prove such an impossibility bound even assuming that when deciding whether $j$ is null, one has the full information of $\theta_{0,\backslash j}$. Then the problem becomes testing a single parameter $\theta_j$ from the effective observation $\theta_j(X_j - \mathbb{E}[X_j|X_{\backslash j}]) + W$ where the signal $X_j - \mathbb{E}[X_j|X_{\backslash j}]$ has variance $[(\mathbf{\Sigma}^{-1})_{jj}]^{-1}$.

Using the fact that $|\{j\colon \hat{\theta}_j \neq 0\}| \leq p$, we obtain from the definition of FDR that $\frac{1}{p}\sum_{j=1}^{p}\mathbb{P}[\theta_j = 0, \hat{\theta}_j \neq 0] \leq q$, and hence

$$\frac{1}{p}\sum_{j=1}^{p}\mathbb{P}[\hat{\theta}_j \neq 0 | \theta_j = 0] \leq \frac{q}{1-\alpha}. \tag{28}$$

By the Markov inequality we have $\frac{1}{q}|\mathcal{S}_1| \leq \sqrt{\frac{q}{1-\alpha}}$ where we defined

$$\mathcal{S}_1 := \left\{ j\colon \mathbb{P}[\hat{\theta}_j \neq 0 | \theta_j = 0] > \sqrt{\frac{q}{1-\alpha}} \right\}. \tag{29}$$

For each $j \in [p] \setminus \mathcal{S}_1$, we have

$$\sqrt{\frac{q}{1-\alpha}} > \mathbb{P}[\hat{\theta}_j \neq 0 | \theta_j = 0] \tag{30}$$

$$= \mathbb{P}\left[\hat{\theta}_j \neq 0 \,\Big|\, \frac{1}{n}\|\bar{\mathbf{X}}_j\|^2 P_{jj} \leq 1.1, \theta_j = 0\right] \cdot \mathbb{P}\left[\frac{1}{n}\|\bar{\mathbf{X}}_j\|^2 P_{jj} \leq 1.1\right] \tag{31}$$

$$\geq \mathbb{P}\left[\hat{\theta}_j \neq 0 \,\Big|\, \frac{1}{n}\|\bar{\mathbf{X}}_j\|^2 P_{jj} \leq 1.1, \theta_j = 0\right] \cdot (1 - o_n(1)) \tag{32}$$

where we defined $\bar{\mathbf{X}}_j := \mathbf{X}_j - \mathbb{E}[\mathbf{X}_j | \mathbf{X}_{\setminus j}]$; (31) used the independence of $\bar{\mathbf{X}}_j$ and $\theta_j$; (32) used the concentration of the $\chi^2$ distribution.

Let us turn to Type-II error:

$$1 - \epsilon \leq \mathbb{E}\left[\frac{|\{j\colon \theta_j \neq 0, \hat{\theta}_j \neq 0\}|}{1 \vee |\{j\colon \theta_j \neq 0\}|}\right] \tag{33}$$

$$\leq \mathbb{E}\left[\frac{|\{j\colon \theta_j \neq 0, \hat{\theta}_j \neq 0\}|}{1 \vee |\{j\colon \theta_j \neq 0\}|} \cdot \mathbb{1}\{|j\colon \theta_j \neq 0| \geq (1-\epsilon)\alpha p\}\right]$$
$$\quad + \mathbb{P}[|j\colon \theta_j \neq 0| < (1-\epsilon)\alpha p] \tag{34}$$

$$\leq \mathbb{E}\left[\frac{|\{j\colon \theta_j \neq 0, \hat{\theta}_j \neq 0\}|}{(1-\epsilon)\alpha p}\right] + \mathbb{P}[|j\colon \theta_j \neq 0| < (1-\epsilon)\alpha p]. \tag{35}$$

Therefore,[4]

$$\frac{1}{p}\sum_{j=1}^{p}\mathbb{P}[\hat{\theta}_j \neq 0 | \theta_j \neq 0] = \frac{1}{\alpha p}\sum_{j=1}^{p}\mathbb{P}[\hat{\theta}_j \neq 0, \theta_j \neq 0] \tag{36}$$

$$\geq (1-\epsilon)(1-\epsilon-o_p(1;\epsilon,\alpha)) \tag{37}$$

$$\geq 1 - 2\epsilon - o_p(1;\epsilon,\alpha). \tag{38}$$

By the Markov inequality, $\frac{1}{p}|\mathcal{S}_2| \leq \sqrt{2\epsilon + o_p(1;\epsilon,\alpha)}$, where we defined

$$\mathcal{S}_2 := \left\{ j\colon \mathbb{P}[\hat{\theta}_j = 0, \theta_j \neq 0] > \sqrt{2\epsilon + o_p(1;\epsilon,\alpha)} \right\}. \tag{39}$$

For $j \in [p] \setminus \mathcal{S}_2$, we have

$$\sqrt{2\epsilon + o_p(1;\epsilon,\alpha)} \geq \mathbb{P}[\hat{\theta}_j = 0, \theta_j \neq 0] \tag{40}$$

$$= \mathbb{P}\left[\hat{\theta}_j = 0 \,\Big|\, \frac{1}{n}\|\bar{\mathbf{X}}_j\|^2 P_{jj} \leq 1.1, \theta_j \neq 0\right] \cdot \mathbb{P}\left[\frac{1}{n}\|\bar{\mathbf{X}}_j\|^2 P_{jj} \leq 1.1\right] \tag{41}$$

$$\geq \mathbb{P}\left[\hat{\theta}_j = 0 \,\Big|\, \frac{1}{n}\|\bar{\mathbf{X}}_j\|^2 P_{jj} \leq 1.1, \theta_j \neq 0\right] \cdot (1 - o_n(1)). \tag{42}$$

Using Neyman-Pearson's lemma, we can lower bound $\mathbb{P}\left[\hat{\theta}_j \neq 0 \,\big|\, \frac{1}{n}\|\bar{\mathbf{X}}_j\|^2 P_{jj} \leq 1.1,\, \theta_j = 0\right] +$

$\mathbb{P}\left[\hat{\theta}_j = 0 \,\big|\, \frac{1}{n}\|\bar{\mathbf{X}}_j\|^2 P_{jj} \leq 1.1,\, \theta_j \neq 0\right]$ by

$$1 - \frac{1}{2}|\mu_1 - \mu_2| \geq 1 - \sqrt{\frac{1}{2}D(\mu_1\|\mu_2)} \tag{43}$$

$$\geq 2Q\left(\frac{1}{2}\sqrt{\frac{1.1}{P_{jj}}}\right) \tag{44}$$

where $\mu_1$ and $\mu_2$ are one dimensional Gaussian distributions with the same variance $n$ but differ in mean by $\sqrt{1.1n/P_{jj}}$. But, for $j \in [p] \setminus (\mathcal{S}_1 \cup \mathcal{S}_2)$, we can upper bound it by

$$\phi := \frac{1}{1 - o_n(1)}\left(\sqrt{\frac{q}{1-\alpha}} + \sqrt{2\epsilon + o_p(1;\epsilon,\alpha)}\right), \tag{45}$$

and hence from (44) and (45),

$$P_{jj} \leq \frac{1.1}{\left(2Q^{-1}(\phi/2)\right)^2}. \tag{46}$$

Thus

$$\|(P_{jj})_{j=1}^p\|_{LP} \leq \max\left\{\frac{1.1}{\left(2Q^{-1}(\phi/2)\right)^2},\, \frac{1}{p}|\mathcal{S}_1 \cup \mathcal{S}_2|\right\} \tag{47}$$

$$\leq \max\left\{\frac{1.1}{\left(2Q^{-1}(\phi/2)\right)^2},\, \frac{q}{1-\alpha} + \sqrt{2\epsilon + o_p(1;\epsilon,\alpha)}\right\}. \tag{48}$$

$\qquad\qquad\qquad\qquad\qquad\qquad\qquad\qquad\qquad\qquad\qquad\qquad\qquad\qquad\qquad\qquad\qquad\qquad\square$

# B  Proof of Proposition 4

*Proof.* The main work is to show that $\tau$ defined in the standard distributional limit is bounded independently of $\boldsymbol{\Sigma}$. Recall from [JM14, (37)] that $\tau$ satisfies the equation

$$\tau^2 = \sigma_0^2 + \frac{1}{\delta}\lim_{p\to\infty}\frac{1}{p}\mathbb{E}[\|\eta_{1/\mathsf{d}}(\theta_0 + \tau\boldsymbol{\Sigma}^{-1/2}\mathbf{Z}) - \theta_0\|_{\boldsymbol{\Sigma}}^2] \tag{49}$$

where $\mathbf{Z} \sim \mathcal{N}(\mathbf{0}, \mathbf{I})$, $\|\mathbf{y}\|_{\boldsymbol{\Sigma}} := \sqrt{\mathbf{y}^\top\boldsymbol{\Sigma}\mathbf{y}}$, $1/\mathsf{d} = 1 - \|\hat{\theta}\|_0/n \geq 1 - 1/\delta$. and the proxy operator is defined by

$$\eta_{1/\mathsf{d}}(\mathbf{y}) := \operatorname{argmin}_{\theta \in \mathbb{R}^p}\left\{\frac{1}{2\mathsf{d}}\|\theta - \mathbf{y}\|_{\boldsymbol{\Sigma}}^2 + \lambda\|\theta\|_1\right\}. \tag{50}$$

We note that the proxy operator $\eta_{1/\mathsf{d}}$ is non-expansive in $\|\cdot\|_{\boldsymbol{\Sigma}}$. Indeed, consider arbitrary $\mathbf{y}_1, \mathbf{y}_2$, and let $\theta_1, \theta_2$ be such that $0 \in \boldsymbol{\Sigma}(\theta_k - \mathbf{y}_k) + \partial\mathcal{L}(\theta_k)$, $k = 1, 2$, where $\partial\mathcal{L}$ denotes the subgradient of the convex functional $\lambda\mathsf{d}\|\cdot\|_1$. We then have

$$\boldsymbol{\Sigma}(\mathbf{y}_1 - \mathbf{y}_2) \in \boldsymbol{\Sigma}(\theta_1 - \theta_2) + \partial\mathcal{L}(\theta_1) - \partial\mathcal{L}(\theta_2), \tag{51}$$

and hence there exist $G_1 \in \partial\mathcal{L}(\theta_1)$ and $G_2 \in \partial\mathcal{L}(\theta_2)$ such that $\|\theta_1 - \theta_2\|_{\boldsymbol{\Sigma}}^2 \leq \|\theta_1 - \theta_2\|_{\boldsymbol{\Sigma}}^2 + \langle G_1 - G_2, \theta_1 - \theta_2\rangle = \langle\theta_1 - \theta_2, \mathbf{y}_1 - \mathbf{y}_2\rangle_{\boldsymbol{\Sigma}} \leq \|\theta_1 - \theta_2\|_{\boldsymbol{\Sigma}} \cdot \|\mathbf{y}_1 - \mathbf{y}_2\|_{\boldsymbol{\Sigma}}$, where we used the convexity of $\mathcal{L}(\cdot)$. This shows the non-expansiveness of $\eta_{1/\mathsf{d}}$. We now upper bound the right side of (49) by noting that

$$\|\eta_{1/\mathsf{d}}(\theta_0 + \tau\boldsymbol{\Sigma}^{-1/2}\mathbf{Z}) - \theta_0\|_{\boldsymbol{\Sigma}}^2$$

$$\leq \frac{1+\delta}{2}\|\eta_{1/\mathsf{d}}(\theta_0 + \tau\boldsymbol{\Sigma}^{-1/2}\mathbf{Z}) - \eta_{1/\mathsf{d}}(\theta_0)\|_{\boldsymbol{\Sigma}}^2 + \frac{\delta+1}{\delta-1}\|\eta_{1/\mathsf{d}}(\theta_0) - \theta_0\|_{\boldsymbol{\Sigma}}^2 \tag{52}$$

$$\leq \frac{1+\delta}{2}\|\tau\boldsymbol{\Sigma}^{-1/2}\mathbf{Z}\|_{\boldsymbol{\Sigma}}^2 + \frac{\delta+1}{\delta-1}\|\eta_{1/\mathsf{d}}(\theta_0) - \theta_0\|_{\boldsymbol{\Sigma}}^2 \tag{53}$$

$$\leq \frac{1+\delta}{2}\tau^2 p + \frac{\delta+1}{\delta-1}\left(\|\eta_{1/\mathsf{d}}(\theta_0) - \theta_0\|_{\boldsymbol{\Sigma}}^2 + 2\mathsf{d}\lambda\|\eta_{1/\mathsf{d}}(\theta_0)\|_1\right) \tag{54}$$

$$\leq \frac{1+\delta}{2}\tau^2 p + \frac{\delta+1}{\delta-1}\cdot 2\lambda\mathsf{d}\|\theta_0\|_1. \tag{55}$$

382 Therefore by (49),

$$\tau^2 \leq \frac{2\delta\sigma_0^2}{\delta - 1} + \frac{2\lambda\delta(\delta + 1)\beta}{(\delta - 1)^3}. \tag{56}$$

383 Suppose that the algorithm selects $j$ such that $|\hat{\theta}_j^u| < t$ as nulls, for some threshold $t \in (0, 1)$. Let
384 $\mathcal{H}_0 := \{j \colon \theta_{0,j} = 0\}$ and $\mathcal{H}_1 := [p] \setminus \mathcal{H}_0$. We have

$$|\{j \in \mathcal{H}_0, |\hat{\theta}_j^u| \geq t\}| \leq \inf_{s>0} \left[ |\{j \in \mathcal{H}_0 \colon |\hat{\theta}_j^u| \geq \tau s P_{jj}^{1/2}\}| + |\{j \in \mathcal{H}_0 \colon \tau s P_{jj}^{1/2} \geq t\}| \right]. \tag{57}$$

385 For fixed $s$ independent of $p$, by the definition of standard distributional limit, we have, almost surely,

$$\limsup_{p\to\infty} \frac{1}{p} |\{j \in \mathcal{H}_0 \colon |\hat{\theta}_j^u| \geq \tau s P_{jj}^{1/2}\}| \leq \mathbb{P}[|Z|\Upsilon^{1/2} \geq s\Upsilon^{1/2}] \tag{58}$$

$$= \mathbb{P}[|Z| \geq s] \tag{59}$$

386 where $Z \sim \mathcal{N}(0, 1)$ and $\Upsilon$ is a random variable whose distribution is the weak limit of the empirical
387 distribution of $(P_{jj})_{j=1}^p$ (from the definition of the standard empirical distribution). Moreover, if
388 $s \leq \frac{t}{\tau\sqrt{L}}$, then

$$\limsup_{p\to\infty} \frac{1}{p} |\{\tau s P_{jj}^{1/2} \geq t\}| \leq \mathbb{P}\left[\Upsilon \geq \left(\frac{t}{s\tau}\right)^2\right] \tag{60}$$

$$\leq L \tag{61}$$

389 almost surely, where $L := \lim_{p\to\infty} \|(P_{jj}^{(p)})_{j=1}^p\|_{LP}$. Substituting into (57), we obtain

$$\limsup_{p\to\infty} \frac{1}{p} |\{j \in \mathcal{H}_0, |\hat{\theta}_j^u| \geq t\}| \leq \mathbb{P}\left[|Z| \geq \frac{t}{\tau\sqrt{L}}\right] + L. \tag{62}$$

390 By the same arguments, we also have

$$\limsup_{p\to\infty} \frac{1}{p} |\{j \in \mathcal{H}_1, |\hat{\theta}_j^u| \leq t\}| \leq \limsup_{p\to\infty} \frac{1}{p} |\{j \in \mathcal{H}_1, |\hat{\theta}_j^u - \theta_{0,j}| \geq 1 - t\}| \tag{63}$$

$$\leq \mathbb{P}\left[|Z| \geq \frac{1-t}{\tau\sqrt{L}}\right] + L \tag{64}$$

391 almost surely. Choosing $t = 1/2$ and noting that $|\mathcal{H}_1|/p \to \alpha$ shows that we can bound

$$\limsup_{p\to\infty} FDR^{(p)} \leq 2Q\left(\frac{1}{2\tau\sqrt{L}}\right) + L \tag{65}$$

392 and

$$\liminf_{p\to\infty} POWER^{(p)} \geq 1 - \frac{1}{\alpha}\left[2Q\left(\frac{1}{2\tau\sqrt{L}}\right) + L\right]. \tag{66}$$

393 $\qquad\qquad\qquad\qquad\qquad\qquad\qquad\qquad\qquad\qquad\qquad\qquad\qquad\qquad\qquad\qquad\qquad\qquad\qquad\qquad\qquad\qquad\quad$ $\square$

## 394 C   Proof of Lemma 5

395 *Proof.* According to the definition of the standard distributional limit, there exists $\tau \neq 0$ such
396 that with probability 1, the empirical distribution of $\{\left((\hat{\underline{\theta}}_j^u - \underline{\theta}_{0,j})/\tau, (\underline{\Sigma}^{-1})_{jj}\right)\}_{j=1}^{2p}$ (which is
397 random since $\mathbf{Y}$ and $\mathbf{X}$ are random) convergences weakly to the distribution of $(\Upsilon^{1/2}Z, \Upsilon)$ where
398 $Z \sim \mathcal{N}(0, 1)$ is independent of $\Upsilon$.

399 Since for any number $t'$, $W_j := |\hat{\underline{\theta}}_j^u| - |\hat{\underline{\theta}}_{j+d}^u| \leq -t'$ implies $|\hat{\underline{\theta}}_{j+d}^u| \geq t'$, by the same steps up to
400 (62), we have

$$\limsup_{p\to\infty} \frac{1}{2p} |\{j \in [p] \colon W_j \leq -t'\}| \leq 2Q(t'/\tau\sqrt{L}) + L. \tag{67}$$

But

$$|\{j \in \mathcal{H}_1 \colon W_j \geq t'\}| \geq |\mathcal{H}_1| - |\{j \in \mathcal{H}_1 \colon W_j \leq t'\}| \tag{68}$$

$$\geq |\mathcal{H}_1| - |\{j \in \mathcal{H}_1 \colon |\hat{\underline{\theta}}_j^u| \leq 2t'\}| - |\{j \in \mathcal{H}_1 \colon |\hat{\underline{\theta}}_{j+d}^u| \geq t'\}| \tag{69}$$

$$\geq |\mathcal{H}_1| - |\{j \in [p] \colon |\hat{\underline{\theta}}_j^u - \underline{\theta}_{0,j}| \geq 1 - 2t'\}| - |\{j \in [p] \colon |\hat{\underline{\theta}}_{j+d}^u| \geq t'\}|, \tag{70}$$

where $\mathcal{H}_1 := \{j \colon \theta_{0,j} \neq 0\}$. Now again using the same steps up to (62), we conclude that almost surely,

$$\liminf_{p \to \infty} \frac{1}{2p} |\{j \in \mathcal{H}_1 \colon W_j \geq t'\}| \geq \alpha/2 - 2Q((1-t')/\tau\sqrt{L}) - 2Q(t'/\tau\sqrt{L}) - 2L. \tag{71}$$

Since

$$T := \min \left\{ t \colon \frac{|\{j \in [p] \colon W_j \leq -t\}|}{|\{j \in [p] \colon W_j \geq t\}| \vee 1} \leq q \right\} \tag{72}$$

and we chose

$$\frac{2Q(t'/\tau\sqrt{L}) + L}{\alpha/2 - 2Q((1-t')/\tau\sqrt{L}) - 2Q(t'/\tau\sqrt{L}) - 2L} < q, \tag{73}$$

we see that almost surely, $T \leq t'$ for $p$ large enough. Thus the number of true positives using the data dependent threshold $T$ is larger than the number of true positives using the threshold $t'$. The claim follows by choosing $t' = 1/3$ and using (71). $\qquad\square$

## D Proofs in Section 6

*Proof of Theorem 6.* From linear algebra, we see that a necessary and sufficient condition such that (27) fulfills the positive semidefinite condition for the joint covariance matrix is that

$$2\operatorname{diag}(\operatorname{diag}(\boldsymbol{\Sigma}^{-1})) - \boldsymbol{\Sigma}^{-1} \succeq 0 \tag{74}$$

In other words, we want the precision matrix to maintain p.s.d. after flipping the signs of the off-diagonals. This is true in the diagonally dominant case.

Using Hammersley theorem we know that the nonzero patter of the precision matrix (inverse of the covariance matrix) corresponds to the connectivity graph of the graphical model, which is a tree in the current case. The claim then follows from Lemma 9 below. $\qquad\square$

**Lemma 9.** $\mathbf{P}$ *is a square matrix and the nonzero pattern of* $\mathbf{P}$ *corresponds to a forest (a union of trees), then* $\mathbf{P}$ *and* $2\operatorname{diag}(\mathbf{P}) - \mathbf{P}$ *have the same set of eigenvalues.*

*Proof.* Assume without loss of generality that the first entry corresponds to a leaf and the second entry corresponds to its unique neighbor. We can expand the determinant to check that the characteristic polynomial satisfies

$$\det(\lambda \mathbf{I}_p - \mathbf{P}) = (\lambda - P_{11}) \det(\lambda \mathbf{I}_{p-1} - P_{[2:n] \times [2:p]}) - P_{12}^2 \det(\lambda \mathbf{I}_{p-2} - P_{[3:p] \times [3:p]}) \tag{75}$$

where $p$ is the size of $\mathbf{P}$, and $P_{[2:p] \times [2:p]}$ denotes the principle submatrix of $\mathbf{P}$ consisting of entries of $\mathbf{P}$ with indices in $\{2, \ldots, p\} \times \{2, \ldots, p\}$. Note that $P_{[2:p] \times [2:p]}$ and $P_{[3:p] \times [3:p]}$ also correspond to forrests. By induction, we see that the off-diagonal coefficients enter the characteristic polynomial only through their squares. In other words, the characteristic polynomial is unchanged after flipping the signs of the off-diagonals. $\qquad\square$

*Proof of Theorem 7.* The $\{1, \ldots, p\} \times \{1, \ldots, p\}$-submatrix of the precision matrix satisfies

$$(\underline{\mathbf{P}}_{[p] \times [p]})^{-1} = 2\operatorname{diag}(\mathbf{s}) - \operatorname{diag}(\mathbf{s})\boldsymbol{\Sigma}^{-1}\operatorname{diag}(\mathbf{s}) \tag{76}$$

$$= 2\operatorname{diag}^{-1}(\mathbf{P}) - \operatorname{diag}^{-1}(\mathbf{P})\mathbf{P}\operatorname{diag}^{-1}(\mathbf{P}) \tag{77}$$

where $s_j = P_{jj}^{-1}$, $j = 1, \ldots, p$ in the case of conditional expectation knockoff. Note that $2\operatorname{diag}^{-1}(\mathbf{P}) - \operatorname{diag}^{-1}(\mathbf{P})\mathbf{P}\operatorname{diag}^{-1}(\mathbf{P})$ and $\operatorname{diag}^{-1}(\mathbf{P})\mathbf{P}\operatorname{diag}^{-1}(\mathbf{P})$ have the same diagonals, but

the off-diagonals are of the opposite signs and equal absolute values. When $\mathbf{P}$ is assumed to be associated with a tree, these two matrices have the same spectral, and in particular, have the same determinant. By the same reasoning, all their principal minors have the same determinant. Therefore the

$$\text{diag}(\underline{\mathbf{P}}_{[p]\times[p]}) = \text{diag}^{-1}(\mathbf{P})\,\text{diag}(\mathbf{P}^{-1})\,\text{diag}^{-1}(\mathbf{P}) \tag{78}$$

$$= \text{diag}^{-1}(\mathbf{P})\,\text{diag}(\mathbf{\Sigma})\,\text{diag}^{-1}(\mathbf{P}). \tag{79}$$

$\square$

*Proof of Theorem 8.* Recall that $(\underline{\mathbf{P}}_{[p]\times[p]})^{-1} = 2\,\text{diag}(\mathbf{s}) - \text{diag}(\mathbf{s})\mathbf{\Sigma}^{-1}\,\text{diag}(\mathbf{s})$. In the case of equi-knockoff, one selects $s_j = \lambda_{min}(\mathbf{\Sigma})$, and we have

$$\lambda_{min}(\mathbf{\Sigma})\mathbf{I} \preceq 2\,\text{diag}(\mathbf{s}) - \text{diag}(\mathbf{s})\mathbf{\Sigma}^{-1}\,\text{diag}(\mathbf{s}) \preceq \lambda_{min}(\mathbf{\Sigma})\mathbf{I}. \tag{80}$$

$\square$

# E   Notes on the experiments

Our code is built upon the knockoff software on Emmanuel Candès's website,

```
https://web.stanford.edu/group/candes/knockoffs/software/knockoffs/
```

with the slight modification that $W_j$ is computed using the unbiased coefficients (see Section 5). We hope to post the details of the changes of the codes and our simulation codes at the time of final submission.

It is worth mentioning that the code chooses the Lasso parameter $\lambda$ via cross validation, whereas our theoretical analysis chooses any $\lambda$ independent of $p$.