[Reviews · NeurIPS 2019]

Reviewer 1



II think there might be good stuff in here, but it was pretty abstract for a neurips paper. A more technical journal might be more appropriate. I believe this is the setup. You have a bunch of iid samples of the form - X_i ~ N(0,Sigma) - Y_i ~ N(theta^T X_i , sigma^2) where there are some j for which theta_j=0 and for all other j we have |theta_j|>1. The task is to estimate the set of indices j such that theta_j=0. To solve this problem: 1) you design a way of making knockoffs (in this case I think that amounts to designing a conditional distribution Xtilde |X so that (X,Xtilde) has all the nice knockoff properties) 2) you solve min [(Y - theta1^T X) + lambda |theta1|_1] 3) you solve min [(Y - theta2^T Xtilde) + lambda |theta2|_1] 4) You sort the indices by |theta1_j| - |theta2_j|, and take the ones which are most impressive (with a carefully chosen threshold) Some things that made me nervous: 1) in Definition 1 it says the amount of noise for each Y_i is actually proportional to the number of samples, n. It is written N_i ~ N(0, nsigma^2), which I don't understand at all. Why would that be? 2) it is never explicitly stated in definition 1 that X_i ~ N(0,Sigma), but I assume this is the case? 3) Any nontrivial example of how to design gaussian knockoffs would be nice. For example, say X is a tree graphical model. How might I make my knockoffs? Apparently "conditional independence knockoff" gives me a way, but I didn't understand how you actually design the distribution. 4) you call your regression estimates "lasso coefficients." That tells me that you must have fit the regression using lasso. But you didn't actually say that, so I was nervous that I might be confused. 5) I can't actually tell whether maybe you fit both lasso's simulatenously, i.e. you fit min [(Y - theta1^T X + theta2^T Xtilde) + lambda |theta1|_1 + lambda |theta2|_1]. 6) I conclude that the nonzero theta_j satisfy |theta_j|>1 because that assumption is made in proposition 4, but I'm not actually sure that it is made throughout the paper. But assuming I got everything right, you then go on to show conditions under which your discovery rate was high. Unfortunately, I couldn't make heads or tails of those conditions. I can read them and understand their literal meaning, but I have no intuition at all. The marginal variances Sigma_{jj} seems to have some very important role, I think I want them small. It almost seems like I need sum_j Sigma_jj to be bounded in the limit as p->infinity with n/p>1 -- but that seems like an absurd assumption, since it would imply that most of the variables are essentially deterministic?

Reviewer 2



Significance: Recently knockoffs have been proposed to control FDR in multiple testing problems. Alot of the literature in this area deals with designing procedures with guaranteed FDR control. However, the power analysis of these procedures is not well understood. This paper tackles the important problem of analyzing the power of knockoff procedures under an idealized setup with correlated gaussian features. Originality: The paper leverages previous AMP-based analysis of the LASSO estimator to derive their theoretical results. Existing work leveraged this analysis to study the power of knockoff procedures with i.i.d. Gaussian design, this work considers correlated gaussian design. Clarity: The work is reasonably well written, but there are still several typos. Detailed comments: Proposition 2,4: Proposition 2,4 provide a tight characterization of the Information Theoretic limit of achieving FDR=0 and POWER = 1 asymptotically. Can the author clarify how is the task of achieving asymptotically FDR = 0 and POWER = 1 different from support recovery? Are any of these results implied by existing results regarding support recovery? Lemma 5: From my understanding, this is only a sufficient condition to guarantee FDR = 0 and POWER = 1 asymptotically. This is a limitation of this work since the comparison between different knockoff procedures is done based on this sufficient condition, and it is not clear that this sufficient condition is tight. Simulations: The authors use simulations to demonstrate that comparision between theoretically derived sufficient condition (ESD) reflects the performance of the knockoff procedures in simulations. In particular the ESD condition predicts the failure of equi-knockoff procedure correctly. However the performance difference between conditional knockoff and sdp-knockoff don't seem to be significant and so these simulations don't really tell us anything about the power of the ESD condition to compare these two techniques. It would be great if the authors could find setups where these two knockoff methods have significantly different performance and compare the ESD in these setups.

Reviewer 3



Originality: Although the result was derived under Gaussian assumption, it is interesting and includes the original result; the authors give the notion of ESD that tends to zero implies type II error (1-power) goes to zero, and show that the power is closely related to the empirical distribution of the precision matrix of regressors. Quality: The theoretical result is constructed using Javanmard and Montanari (2014, IEEE-T-IT) and looks fine. Clarity: The paper focuses mainly on the theory, but it seems better to add some comments on the connection to practical viewpoints. For example, can many real data satisfy the conditions of Proposition 4? Significance: It is significant as there has been no such formal theoretical analysis on power in knockoffs inference. It is also important for empirical researchers since it is used for selecting a knockoff that yields higher power.

[Author Response · NeurIPS 2019]

We cordially thank the reviewers for their time and thoughtful comments. We will improve the presentation of the paper
and further develop the experiments, including testing on real data (which are abundant in the prior works on LASSO),
as suggested. Moreover, some specific comments by the reviewers are addressed as follows:

**Reviewer 1**

- Regarding the appropriateness of our paper to neurips: We would like to mention that there have been previous
papers in the same category which appeared on neurips (and were highly influential). For example, The
Javanmard and Montanari paper we cited has a neurips 2013 version: "Confidence Intervals and Hypothesis
Testing for High-Dimensional Statistical Models". We have plans of further developing the experiments, and
submitting a longer version to journal for reference. Thank you for the suggestions.

- In Definition 1, indeed the noise scales with the number of samples: $N_i \sim \mathcal{N}(0, n\sigma^2)$. This scaling ensures
that for each $j$, the square error in estimating $\theta_j$ scales as $\Theta(n)/n = \Theta(1)$ (where $1/n$ factor because of $n$
samples), in which case we have hope of convergence of the empirical distribution of the error, as we desire.

- In Definition 1, indeed $\mathbf{X}_i \sim \mathcal{N}(\mathbf{0}, \mathbf{\Sigma})$. We updated the manuscript.

- Example of constructing Gaussian knockoffs: in the Gaussian case it all boils down to designing the covariance
matrix. eq 5-10 show the previous ways, and for conditional independence knockoff, the explict formula is
given in (27).

- By "regressing $Y$ on $[X^p, \tilde{X}^p]$", we mean solving $\min_{\theta=(\theta_1,\theta_2)}\{\|Y - [X^p, \tilde{X}^p]\theta\|^2 + \lambda\|\theta\|_1\}$, as opposed to
solving $\min_{\theta_1}\{\|Y - X^p\theta_1\|^2 + \lambda\|\theta_1\|_1\}$ and $\min_{\theta_2}\{\|Y - \tilde{X}^p\theta_2\|^2 + \lambda\|\theta_2\|_1\}$ separately. We will further
clarify this in the texts around (3).

- Assumption of $|\theta_j| \geq 1$: actually we can completely drop this assumption, using a more careful analysis, and
the bounds (22) and (23) will be replaced by new bounds depending on the limit of the empirical distribution
of $\theta_0^{(p)}$ (whose existence is guaranteed by the standard distributional limit assumption).

- Simply put, we found a simple necessary and sufficient condition on $\mathbf{\Sigma}$ for low FDR and high power: empirical
distribution of $((\mathbf{\Sigma}^{-1})_{jj})_{j\in[p]}$ should converge to 0 in distribution. For example, since convergence in
expectation implies convergence in distribution (Markov inequality), having $\frac{1}{p}\sum_{j=1}^{p}(\mathbf{\Sigma}^{-1})_{jj}$ bounded above
is sufficient.

**Reviewer 2**

- Regarding the necessity of the condition for FDR$\to 0$ and POWER$\to 1$ for the knockoff filter: actually
our condition $\|(\underline{P}_{jj})_{j=1}^{2p}\|_{LP} \to 0$ is not only sufficient but also necessary. The intuitive explanation is that
$\hat{\theta}_j^u - \theta_{0,j}$ is roughly distributed as $\mathcal{N}(0, \tau^2\underline{P}_{jj}^{-1})$ (by the standard distributional limit), so that FDR$\to 0$ and
POWER$\to 1$ if and only if the fraction of $j$'s for which $\underline{P}_{jj}$ exceeds any given threshold asymptotically vanish
(i.e., weak convergence of the empirical distribution of $(\underline{P}_{jj})_{j\in[2p]}$). The proof of converse will be similar to
achievability. However, the reviewer is right that the we should make this point clearer in the revised version.

- Support recovery: the variable selection problem might be interpreted as support recovery. However, to our
knowledge, literature on support recovery usually focuses on exactly recovering the whole support (e.g. Knight
and Fu, "Asymptotics for lasso-type estimators," 2000, and Zhao and Yu, "On model selection consistency of
Lasso," 2006). In contrast, FDR may be considered as a softer criterion for the quality of support recovery.
Exact support recovery is not asymptotically feasible in the regime we consider. We will add discussions and
related citations.

- Comparison with SDP knockoff: testing binary tree with various correlations yield similar simulation results,
namely that the conditional independence knockoff performs similarly but slightly better than sdp knockoff.
We plan to test other trees or sparse graphs. Notwithstanding, the conditional independence knockoff is still
much more computationally efficient to construct than sdp knockoff, and more reliable since ESD has a
closed-form expression.

**Reviewer 3**

- Regarding relevance to real data: assumptions such as sparse precision matrix or tree graphical models are
very common, and in fact, many algorithms (such as Chow-Liu or Graphical Lasso) rely on such assumptions
for any hope of estimating of the precision matrix. Also as noted above, we can drop condition (16) in Prop 4
and use the limiting empirical distribution of $\theta_0$ instead. We will further test the algorithm in certain real data
sets. A starting point might be similar data sets in the previous Lasso literature, such as those in the paper of
Javanmard and Montanari. For sparse precision matrix, we are considering similar data as those found in the
papers of Bühlmann, Kalisch, and Meier.

[Meta-Review · NeurIPS 2019]

The authors study knockoffs and derive results on asymptotic FDR and Power in a variety of settings. Despite the fact that the assumptions are quite strong in some regards, the reviewers were generally positive and I am as well. Some reviewers struggled understanding parts of the paper and described ways in which the presentation could be improved; I would like the authors to address these the best they can.